# Ecological drivers of ultraviolet colour evolution in snakes

Hayley L. Crowell [1,4], John David Curlis [1,4], Hannah I. Weller [2,3,4] & Alison R. Davis Rabosky [1,4] ✉

Ultraviolet (UV) colour patterns invisible to humans are widespread in nature. However, research bias favouring species with conspicuous colours under sexual selection can limit our assessment of other ecological drivers of UV colour, like interactions between predators and prey. Here we demonstrate widespread UV colouration across Western Hemisphere snakes and find stronger support for a predator defence function than for reproduction. We find that UV colouration has evolved repeatedly in species with ecologies most sensitive to bird predation, with no sexual dichromatism at any life stage. By modelling visual systems of potential predators, we find that snake conspicuousness correlates with UV colouration and predator cone number, providing a plausible mechanism for selection. Our results suggest that UV reflectance should not be assumed absent in "cryptically coloured" animals, as signalling beyond human visual capacities may be a key outcome of species interactions in many taxa for which UV colour is likely underreported.

Humans are a poor proxy for the visual capabilities of animals across the tree of life[1]. Colouration and its perception are major factors influencing an organism's survival and reproductive success, yet the limitations of human vision have largely restricted the study to species that are brightly coloured in "visible" wavelengths of light (400–700 nm[2,3]). However, the ability to perceive ultraviolet (UV) wavelengths (300–400 nm) is found in many other animals, and UV colour is used frequently in visual communication within and across species[3,4]. This mismatch between human and animal visual systems can result in cryptic diversity in colouration and create a profound barrier to understanding the ecology of many species[5].

For most systems in which UV colouration is well understood, its primary reported function is reproduction rather than survival benefits. From the role of nectar guides in increasing flower pollination[6] to ultraviolet facial markings for social signalling in fish[7] and birds[8], UV colouration provides iconic examples of how reproduction shapes receiver-dependent signal evolution across taxa[9]. However, any bias

toward studies on reproduction-related mechanisms might not accurately reflect the true distribution of function in nature. Instead, a bias could arise because colour patterns used primarily for crypsis or concealment have been understudied for UV colouration. In theory, multiple types of selection can affect colour evolution, and these processes are hypothesised to produce different signatures on the standing variation of UV colouration within and among species[10].

Here we use a broad-scale survey of snake colouration and visual receiver modelling to test the relative roles of survival and reproduction in UV colour evolution across Western Hemisphere snakes (N = 110 species). Snakes have been previously studied for human-visible colour variation in both crypsis and mimicry[11,12], and they are known to have the visual capacity to perceive UV wavelengths[13]. Even though snakes are nested within the larger "lizard" clade that is known to use UV colouration widely for sexual signalling[14,15], the function and prevalence of UV reflectance remain unreported across snakes. If UV colouration is also under strong sexual selection in snakes, differences should be

[1]Department of Ecology and Evolutionary Biology and Museum of Zoology (UMMZ), University of Michigan, Ann Arbor, MI 48109, USA. [2]Department of Ecology, Evolution and Organismal Biology, Brown University, Providence, RI 02912, USA. [3]Helsinki Institute of Life Sciences, University of Helsinki, Helsinki, Uusimaa 00790, Finland. [4]These authors contributed equally: Hayley L. Crowell, John David Curlis, Hannah I. Weller, Alison R. Davis Rabosky. ✉e-mail: ardr@umich.edu

greatest within species between males and females and most pronounced in reproductive (adult) life stages. Alternatively, natural selection from UV-sighted predators should instead produce differences in UV reflectance among species, predictable by habitat specialisation and activity patterns. Predator-driven evolution could also produce differences between adults and juveniles, but in the opposite direction than predicted by sexual selection, as protective colouration often confers a benefit at vulnerable early life stages (as seen in juvenile lizards[16]). Lastly, if UV colouration is selectively neutral, we would expect a strong phylogenetic signal across taxa and no independent influence of sex, life stage, activity pattern, or habitat. Repeated convergent evolution of UV colouration in response to independent transitions to the same habitat would strongly reject a neutral mechanism.

## Results

### Previous research on UV colouration is skewed towards reproduction

UV coloration has been studied over many years, but this research attention may not be uniformly distributed across functional contexts and clades. To quantify the potential presence and magnitude of bias in reports of UV colour, we systematically reviewed the published literature ($N = 2401$ identified studies; Supplementary Fig. 1, Supplementary Data 1) for function and perception of UV colour. We found that when the function was directly tested or inferred ($N = 281$ studies), reproductive functions like mate choice and pollination were indeed reported much more frequently than survival benefits (Fig. 1a, b, red vs. grey wedges). Across studies of all taxa (Fig. 1a), we found that flowering plants, butterflies, and vertebrates were the primary targets of research to the exclusion of most organismal diversity. Within vertebrates (Fig. 1b), the majority of studies were conducted on birds, with nearly 1/3 of those on only three species from just two families (Great Tits, *Parus major*; Eurasian Blue Tits, *Cyanistes caeruleus*; and King Penguins, *Aptenodytes patagonicus*). We found an additional 230 studies that did not test function but which overwhelmingly (97%) report the widespread ability to perceive UV colour in species across many clades (Supplementary Data 1). Together, these results suggest there is indeed high potential for overlooked UV colouration that would directly affect our understanding of both its function and evolution across systems.

### UV colouration is widespread across snake species

By conducting broad surveys of snake colouration in natural populations using cameras specifically designed to capture both human-visible and UV reflectance under natural lighting, we found that UV colouration is unexpectedly prevalent across snakes (Fig. 2; Supplementary Data 2). Although there was substantial variation in the intensity and physical location of UV reflectance across species (Fig. 2a, Supplementary Figs. 2–4), the overwhelming majority (90%) of species reflected UV wavelengths on some part of their bodies, especially the ventral surface and in the colour white (Fig. 2b; Supplementary Fig. 4). UV reflectance on both dorsal and ventral surfaces showed little phylogenetic signal, even when accounting for phylogenetic uncertainty (all Pagel's lambda < 0.2 and all $p > 0.1$, and thus not significantly different from zero). We also found that UV reflectance on one body surface did not reliably predict reflectance on the other surface (all $R^2 = 0$–0.11). Dorsal UV reflectance was both lower overall and more concentrated into patches along the body than ventral reflectance, especially on heads and chins ($\chi^2_{(5)} = 40.206$; $p < 0.001$). Additionally, both the spectrometer and photographic analyses suggest that human-visible and UV elements are decoupled, as the presence of UV reflectance could not be reliably predicted from a snake's visible colouration alone (Figs. 2c and 3a; Supplementary Figs. 2–4; see Supplementary Note 1 for discussion of scale properties).

### Sex does not predict UV colouration, but age class does

To test for age and sex differences in UV reflectance, we analysed a subset of 101 species for which we had robust within-clade sampling ($N = 410$ individuals). Many species showed significant intraspecific variation in UV reflectance, including between individuals that looked similar in the human-visible spectrum (Fig. 3a, discrete UV polymorphism in Amazonian tree boas, *Corallus hortulanus*). Using mixed-effects models controlling for clade as a random effect, we found no differences in overall, dorsal, or ventral UV reflectance between males and females (all $F < 0.05$; $p > 0.8$; Supplementary Fig. 5). However, we found highly significant differences between age classes, such that juvenile snakes had higher reflectance than adults in all comparisons (all $F > 10.77$; $p < 0.002$; Fig. 3b, c, Supplementary Fig. S6). Together, these findings suggest that sexual selection is unlikely to explain the

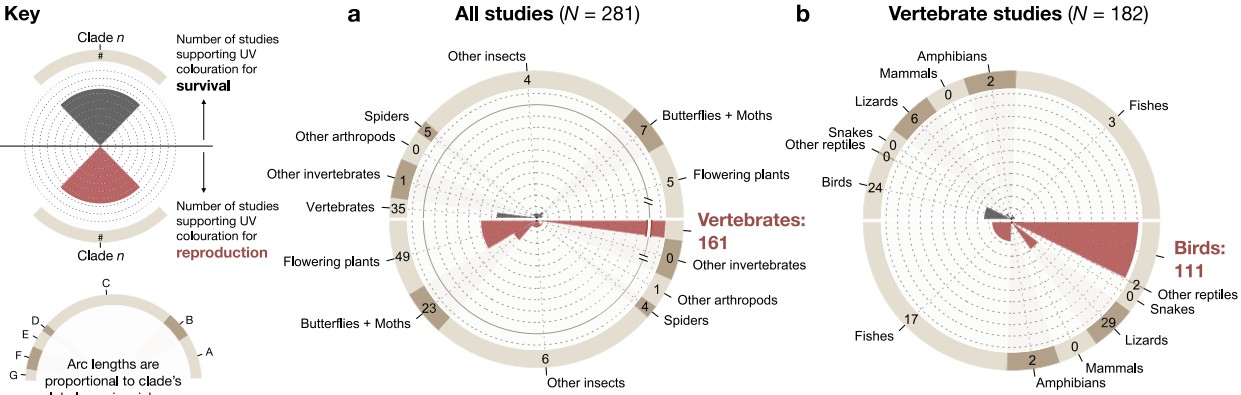

**Fig. 1 | Previous research on ultraviolet colouration predominately targets mating systems.** A literature review of 511 studies shows that when an ecological function for ultraviolet (UV) colouration is tested, reproductive functions like sexual selection and pollination (red wedges) are reported much more frequently than survival benefits (grey wedges) and that they are only tested within a narrow breadth of species. Each dotted ring interval indicates 10 studies. **a** Across all clades, UV colouration is primarily studied as nectar guides in flowering plants and mating systems in butterflies and vertebrates, to the exclusion of most animal diversity. Note the broken axis to accommodate the large number of studies for vertebrates. **b** Within vertebrates, most studies are conducted on bird species (32% of those on only 3 species out of 10,900 described), and most within the context of mating systems. An additional 230 studies that did not test function are not plotted, but these overwhelmingly (97%) report the widespread ability to perceive UV colour in species across all of these clades (see Supplementary Data 1). Snakes are one of only two clades in this analysis with no studies on the function of UV colouration. Source data are provided as Supplementary Data 1.

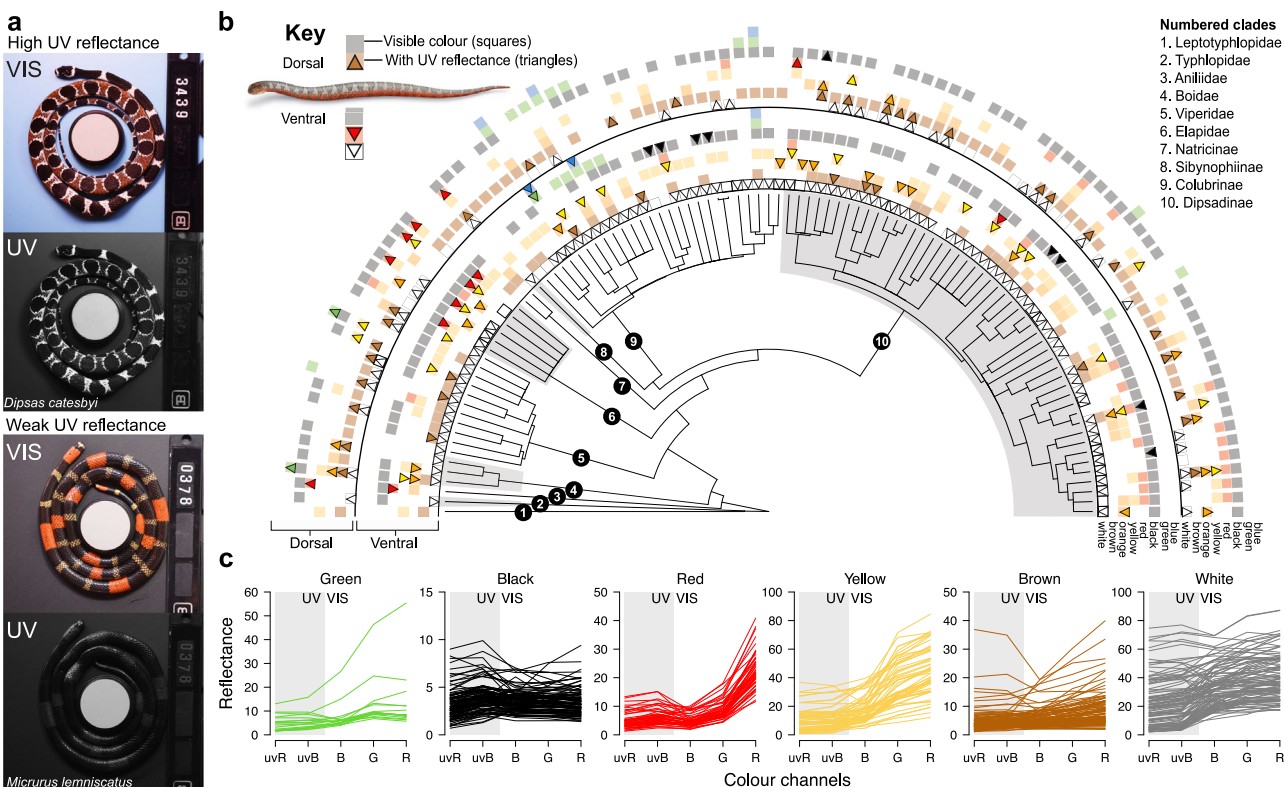

**Fig. 2 | Colour reflectance in the ultraviolet spectrum is widespread across snake species. a** Photographs of snakes taken with multispectral imaging reveal UV reflectance in wavelengths not visible to the human eye. Snakes vary from highly reflective patterns (top) to weak or no UV reflectance (bottom; VIS human-visible wavelengths). **b** Snake species (*N* = 110 measured) vary widely in which visible colours (squares) also reflect in UV wavelengths (triangles), especially across the dorsal side (outer rings) versus the ventral underside of the snake (inner rings). Scientific illustration of *Helicops angulatus* in the graphical legend courtesy of J. Megahan. **c** Snake colour patch measurements (single lines) show that UV reflectance (uvR and uvB values) is not easily predictable from snake visible colouration (RGB values) but is generally highest in white patches (see also Supplementary Figs. 2–4). Source data are provided as a Source Data file.

evolution of UV colouration in snakes, in contrast to many other reptiles[14,15] and other vertebrates (Fig. 1b).

## Habitat ecology predicts UV colour evolution across species

Given that exposure to predators varies depending on the habitat a prey species uses and its relative risk when asleep or awake[17], we also tested whether primary habitat type and diel activity patterns were significant predictors of UV reflectance (Supplementary Data 3). Using phylogenetic generalised linear models, we found that habitat is a significant predictor of dorsal UV reflectiveness ($t$ = 1.75; $p$ = 0.035), with arboreal snakes having significantly higher UV reflectance relative to snakes from all other habitat types (Fig. 3c; Supplementary Fig. 7). Nocturnal snakes also had significantly higher UV reflectance than diurnal snakes ($t$ = 2.05; $p$ = 0.046), and these differences were more extreme for dorsal colour alone (Supplementary Fig. 7). This repeated evolutionary convergence across the same habitat specialisations is inconsistent with a conclusion that neutral processes are driving the evolution of UV reflectance across these snake species, suggesting instead that UV colour is under natural selection.

## Snake conspicuousness to other species depends on UV colouration and chromacy

The three main classes of predators on snakes are birds, mammals, and other snakes[17], which all vary in their ability to perceive different wavelengths, including UV[18,19] (Fig. 4a). We tested the impact of UV reflectance on the conspicuousness of snakes to six different vertebrate receivers as modelled by their measured visual sensitivities to colour[20,21]. We found that snake colour conspicuousness varies greatly among these receivers, with human perceptions of mean dorsal colour

contrast ($\overline{\Delta S}$) being more similar to the dichromatic visual system of other mammals, like dogs, than to the UV-sensitive birds, lizards, and snakes (Fig. 4b; human mean represented by dashed line). By a considerable margin, birds were the receivers for which conspicuousness was the highest. These results are clear even for dorsal surfaces (Fig. 4b), which are maximally visible to predators but which also show reduced UV reflectivity relative to venters (Fig. 3b). Remarkably, up to 30% of the differences in snake conspicuousness among receivers could be explained by the snake's dorsal UV reflectance alone, even though these receivers differ in many aspects of their colour sensitivity (Fig. 4c, d; Supplementary Figs. 8, 9). To test whether the higher colour contrast for UV-sensitive visual systems was only the result of bird visual systems' higher chromacy, we fit a multiple regression model predicting colour contrast as a function of chromacy (2, 3, or 4 cones) and UV sensitivity ($\lambda_{max}$ for the shortest peak sensitivity of any cone). Both UV sensitivity and chromacy were significant predictors of colour contrast (overall model: $F_{(3, 1634)}$ = 409.1, $p$ < 0.001), indicating that the importance of UV reflectance in our results is not solely a consequence of our choice of visual systems. While the relationship between $\Delta S$ and conspicuousness may not be strictly linear[22], our overall results suggest that predation—especially by birds—is a plausible mechanism for the evolution and maintenance of widespread UV coloration in snakes.

## Discussion

This work represents a shift in how the evolution of UV colouration may be understood in animal systems. First, our findings show the highest UV reflectance in juveniles regardless of sex, a result that is far better predicted by a predator-driven hypothesis than by sexual selection alone. These results run counter to most published examples

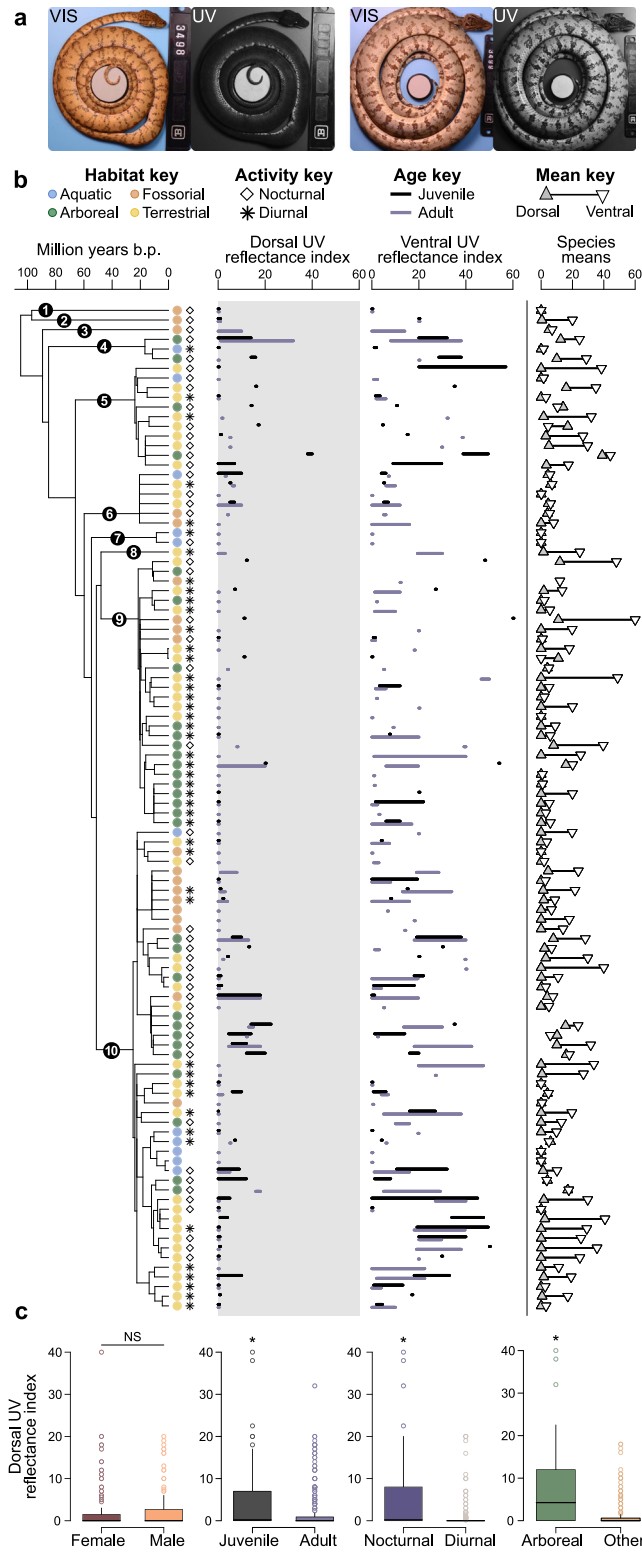

**Fig. 3 | Variation in UV reflectance within and among species is structured by ecology and life stage rather than by sex. a** Individuals within the same species with similar visible colours can vary markedly in UV reflectance, as illustrated here by two Amazonian tree boas from the same population. **b** Total UV reflectance varies within and among species on both the dorsal and ventral surfaces of snakes (*N* = 438 individuals; species means, far right). **c** Within species, juveniles have significantly higher dorsal reflectance than adults ($F_{(1,394)}$ = 10.15, *p* = 0.002), while males and females show no differences in dorsal UV reflectance at any life stage (*N* = 410 individuals; $F_{(1,312)}$ = 0.16, *p* = 0.694). Among species, arboreal and nocturnal species have significantly higher dorsal reflectance than any other ecological category (*N* = 438 individuals; habitat *t* = 1.75, *p* = 0.035; diel activity *t* = 2.05, *p* = 0.046). Statistical tests for sex and age: mixed-effect models with clade as a random effect. Statistical tests for habitat and diel activity: phylogenetic generalised linear models. All tests were two-tailed. See Supplementary Figs. 5–7 for further plots of sex and full ecological data. Asterisks indicate significant differences among factors. For all box plots, centre line represents data median, bounds represent ± 1.5 IQR, and whiskers represent minima and maxima excluding outliers. Source data are provided as a Source Data file.

we cannot exclude snakes as direct receivers of UV signals from other snakes given their capacity for UV-sensitive vision (Fig. 4a), despite colour signals being historically presumed unimportant in communication with other snakes[27]. However, the lack of sexual dichromatism in UV colouration combined with a bias towards high UV in juveniles strongly suggests that potential snake receivers would be functioning in predator rather than mating roles[28]. While there are other ecological factors that can covary with habitat usage (such as diet[29]), we propose that interaction between snakes and UV-sensitive predators is one of the few that could consistently explain our results both within and among species. Regardless of the cellular mechanisms[14] or physical scale properties[30] that may underlie the production of UV reflectance in snakes (which are not fully known, see Supplementary Note 1), the unavoidable consequence of this widespread UV colour is that it is fully visible to selection by many avian predators in the same way as classically studied "visible" snake colours, and it should be considered similarly.

However, we caution that there are multiple ways that species interactions relevant to survival could influence UV colour evolution in snakes. Our results help constrain the space of candidate models by rejecting one class of potential mechanisms, but the distribution of UV reflectance by ecology and life stage could be explained by several non-mutually exclusive alternatives. In particular, we note that having high UV reflectance does not require that this colour functions as a "warning signal" of toxicity to avian predators[31]. UV colour is phylogenetically widespread and clearly not restricted to the two dangerously venomous families in our dataset (Viperidae and Elapidae) or their colour pattern mimics (primarily Dipsadines[12]). Thus, we view a warning colour function as less plausible than two alternative hypotheses that frequently promote lower predation rates across all taxa: (1) passive functions like crypsis while stationary and (2) active defences after predator encounters of sleeping prey.

First, UV colouration could function to reduce detection by diurnal predators while arboreal, nocturnal snakes sleep, as both plants[32] and their epiphytes[33] can have UV reflectance. In this case, UV patterning could make a snake harder to detect by UV-sensitive birds living in trees, as in peppered moths on crustose lichens[34]. In addition, higher amounts of UV reflectance on the ventral surfaces of these snakes could aid in crypsis when viewed from below by terrestrial or understory-dwelling predators, as these colours would presumably contrast less with the UV-intense backdrop of the sky. Nearly all arboreal snakes, both diurnal and nocturnal, are countershaded in the visible spectrum, offsetting lighter bellies with darker dorsal patterning. While some elements of these human-visible patterns are likely neutral, both green patterns (e.g., Supplementary Fig. 2e) and blotchy patterns (e.g., Fig. 2a) are so universal and repeatable in arboreal taxa

of UV colouration (Fig. 1), which usually find the highest UV reflectance in adults or minimal differences across life stages[23–25], and they challenge inferences about the global importance of reproduction in the evolution of UV colour signals. Second, our results expose a neglected axis of colour variation critical to understanding the ecology of snakes. Many "conspicuous" snakes, like the famous red-black banded coral snakes[26], show greatly reduced UV reflectance, while some "cryptic" snakes reveal dramatic intraspecific polymorphism in UV reflectance that is invisible to humans (Fig. 3a, *Corallus hortulanus*). Importantly,

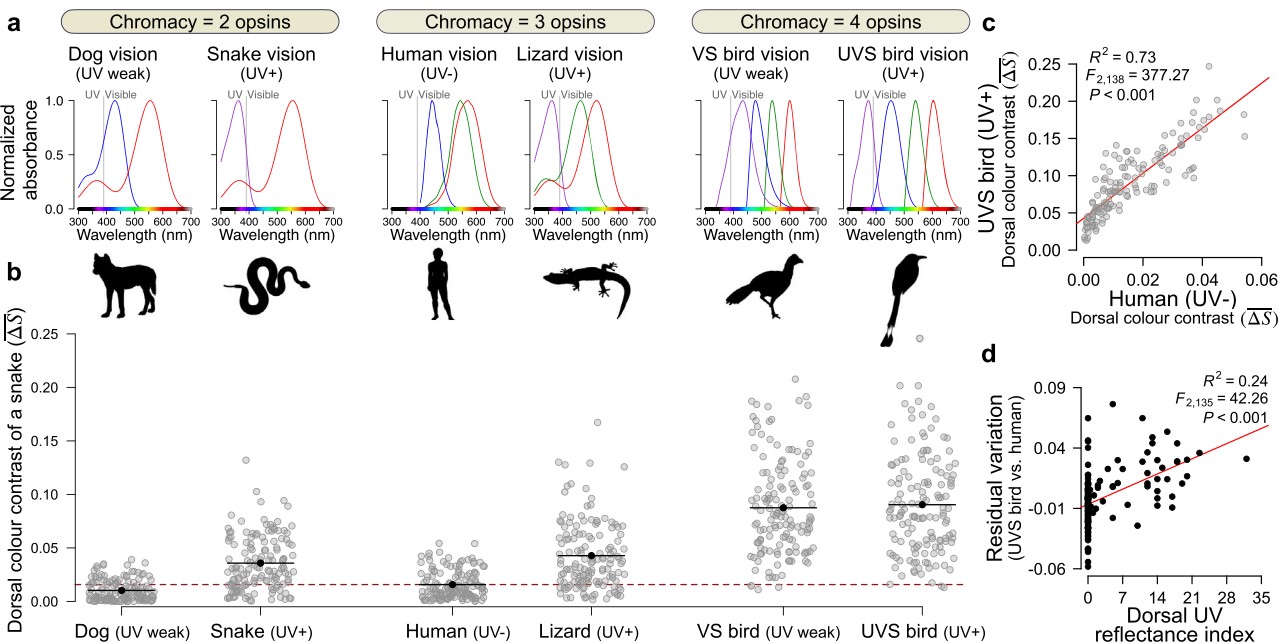

**Fig. 4 | Snake colour conspicuousness varies among observer species and scales with UV perception and chromacy. a** Vertebrate species that interact with snakes in nature have highly variable sensitivities to different wavelengths of light, with different numbers of opsins (absorption curves) and levels of UV sensitivity (<380 nm; VS violet sensitive, UVS ultraviolet sensitive). **b** The mean dorsal colour contrast of individual snakes (a measure of conspicuousness, $\overline{\Delta S}$; $N = 144$ individuals; means indicated by barbels) varies across receivers, scaling with both UV perceptive capacity and chromacy level. Human mean is represented by a dashed line. **c** Comparing the dorsal conspicuousness of each snake (points) for a UVS bird versus a UV- human generates a measure of residual variation in which snakes appear more or less conspicuous to the bird receiver ($p < 0.001$). **d** Up to 30% of this residual variation among these receivers is explained by a snake's UV reflectance alone (24% in this panel, $p < 0.001$; see $R^2$ value and Supplementary Figs. 8, 9 for other receiver comparisons). See Supplementary Table 1 for details and citations of visual systems used, including $\lambda_{max}$, oil droplets/ocular media affecting UV transmission, and calculation of luminance channels. All tests in **c** and **d** were two-tailed. All silhouettes are open access from phylopic.org, with attribution to Yannick Wieger for the violet-sensitive bird. Source data are provided as a Source Data file.

that "avoidance of detection" while exposed on tree branches is widely assumed to underlie the maintenance of such colour patterns[35]. Thus, the additional UV reflectance that we have documented in these otherwise broadly visible pattern elements should not be assumed a priori to function differently.

Second, UV patterns could function as a sort of startle or "signal boosting" colouration displayed by abruptly-awoken snakes to aid in escape immediately following predator encounters of sleeping snakes in a well-lit daytime environment[36–38]. High UV reflectance has been found in other nocturnal lineages, especially within insect groups like moths, which also have higher reflectance on hindwings that are hidden at rest but deployed during diurnal disturbance[39] in the same manner as snake venters[11]. The higher UV reflectance both in juveniles and on the highly visible heads and chins of snakes is also consistent with an active predator defence function because of its similarity to known patterns of visible colouration. In snake species with ontogenetic shifts in colouration, juveniles are almost always more patterned and colourful than adults, whereas the opposite pattern (dull juveniles that become highly patterned adults) is rare to absent[40]. In most cases, this brighter juvenile colouration functions in an anti-predator capacity to protect this sensitive life stage, and juvenile snakes often display these colours more vigorously through increased defensive behaviour[41].

Our data cannot confirm that selection on UV colouration is directional *towards* higher UV reflectance, whether broadly on ventral surfaces or on patched dorsal surfaces on nocturnal, arboreal snakes. An alternative explanation is that dorsal UV reflectance is selected *against* in diurnal and terrestrial snakes, such that UV reflectance is only retained when it is selectively neutral, as on a rarely visible surface like a belly. We note that these mechanisms are also not mutually exclusive, and both processes could interact simultaneously

to produce a snake's full-colour phenotype. However, the latter mechanism of selection against UV colouration would be a wholesale departure from the way it is currently conceptualised in both animal and plant systems, and it would require further experimental testing. Overall, the pervasive restriction of UV reflectance to patches or distinct body regions on the dorsum suggests that UV reflectance on low-visibility ventral surfaces may simply be under weaker selection (as in moths[39]) and that selection on the dorsum is not only strong but in the direction of signal stabilisation rather than diversification (as in tropical birds[42]). This result would run counter to most theories on the evolution of "hidden channel" communication, in which UV signals are expected to have reliable species-specific information that should generate increased signal diversity[4], but would be entirely consistent with crypsis. Experimental tests involving reflectance measurements across the complex background conditions found in nature (e.g., with native lichens, fungi, etc., while in situ on tree branches), combined with predator responses to many combinations of snake colour patterns across these backgrounds, are the critical next step for determining the relative importance of these mechanisms across snake species.

One fundamental insight from this work is that the conspicuousness of an organism's visible colour pattern should not be the primary motivator for choosing which species are studied for UV colouration. The combination of our results and the taxonomic bias visible in Fig. 1 suggest many promising future research targets in other species that experience heavy avian predation, especially insect clades that are arboreal and/or nocturnal, with special attention paid to larval stages that are often less well represented in both research and natural history collections[43]. Ultraviolet signals that evolve for reproduction[5,14,24] may be governed by different evolutionary dynamics than predator-mediated UV reflectance, and new discoveries of UV reflectance should

be tested across these classes of possible drivers to begin assessing these differences[9]. Most importantly, UV reflectance must be directly measured rather than assumed absent or unimportant, especially in systems with minimal or drab visible colour variation[4,11,39] in which UV colour presence and prevalence are likely to be profoundly underreported.

## Methods

### Literature review

To assess trends and biases in the UV literature, we used the Scopus online citation database (accessed 2 November 2023) to identify studies that tested the function of UV colouration across the tree of life (Supplementary Fig. 1). To sample the literature inclusively, we searched within article title, abstract, and keywords using the following set of terms, Boolean operators, and wildcards: ultraviolet OR uv OR "nectar guide" AND (reflect* OR absorb* OR colour* OR colour* OR pattern* OR signal*) AND ("natural selection" OR "sexual selection" OR adapt*). We then restricted this search to only include articles and reviews in the subject areas of (1) biochemistry, genetics, and molecular biology, (2) agricultural and biological sciences, and (3) environmental science. The resulting database included 2401 published manuscripts requiring additional filtering for relevance before scoring for tests of UV function. First, we reviewed titles to remove any articles that were not relevant to UV reflection in an ecological context, such as papers that used UV light in laboratory methodologies, UV radiation in the context of atmospheric science, and human medical treatment ($N = 1516$ removed). We then reviewed the abstract and keywords of each remaining paper to identify the study organisms and the specific biological mechanism for which UV reflection was discussed, with "mechanism" referring to the presence of UV colour on the study organism or the ability to perceive UV wavelengths. If the specific function of UV colour could not be determined from the abstract, we then reviewed the full text of the article. In 362 cases, this additional review of the abstract and text found that the paper was not relevant to UV colouration in an ecological context and thus noted as "NA" for mechanism.

We then classified scored mechanisms within categories of "survival" ($N = 38$), "reproduction" ($N = 224$), "both" ($N = 20$), "unknown" ($N = 40$, generally reporting only the existence of UV colour), or "other" ($N = 201$), with the last category overwhelmingly pertaining to UV perception (97%). Papers grouped into the "survival" category included mechanisms such as camouflage, warning or startle colouration, assistance with prey acquisition, and protection from UV radiation via reflectance. Papers assigned to the "reproduction" category included intraspecific signalling, sexual selection, and pollination. Most tests of intraspecific communication focused on signals relevant to reproduction (mate choice, honesty of signals, access to breeding sites, etc.) and did not explicitly test for dual functions related to survival, but we scored some of these studies as "both" if survival was directly addressed. Papers focused on perception or visual tuning were placed into the "other" category because they did not discuss a specific function of UV reflectance. Papers that discussed only UV fluorescence or absorption (rather than reflection) were scored as "NA" for mechanism, as neither of these phenomena represents true UV reflective colouration. Three authors (HLC, JDC, and ADR) contributed to the first pass on these scores, and then one author (ADR) performed quality control and standardisation to ensure uniformity among scorers, with disagreements discussed until consensus. After all filtering and scoring steps, 523 articles that tested for the function or perception of UV colouration were included in our assessment of relevant previous research (511 of which mapped to taxonomic categories in Fig. 1; see Supplementary Fig. 1and Supplementary Data 1).

### Field collection

We captured 438 snakes from field expeditions to Peru (2016–2018), Nicaragua (2018), Belize (2019), and the United States (Texas,

Colorado; 2021). We collected snakes through a combination of opportunistic foot surveys, drift-fence lines with pitfall and funnel traps, and driving surveys[44]. All field methods were approved by Institutional Animal Care and Use Committees (the University of Michigan #PRO00006234 and #PRO00008306, Dickinson College #895) and respective governmental authorities (Peru: Servicio Nacional Forestal y de Fauna Silvestre 029-2016-SERFOR-DGGSPFFS, 405-2016-SERFOR-DGGSPFFS, 116-2017-SERFOR-DGGSPFFS; Nicaragua: Ministerio del Ambiente y los Recursos Naturales DGB-IC-058-2017, DGPNB-IC-019-2018, DGPNB- IC-020-2018, DGPNB-IC-002-2019; Belize: Forest Department of the Ministry of Agriculture, Fisheries, Forestry, the Environment and Sustainable Development FD/WL/1/19(10); Texas Parks and Wildlife, #SPR-1020-175; Colorado Department of Natural Resources, #1950298916). Although we captured and released several live specimens, the majority were vouchered for natural history museums, with specimen numbers and collections of deposition given in Supplementary Data 1.

### Photography

To quantify reflectance at both visible and UV wavelengths, we used a Nikon D7000 DSLR camera (Nikon Inc., Melville, NY, USA) with full-spectrum conversion (LifePixel; Mukilteo, USA) and equipped with a Coastal Optics UV–VIS–IR 60 mm F/4 macro lens (Jenoptik Optical Systems, Jupiter, FL, USA). We photographed the dorsal and ventral surfaces of each specimen with both a Baader (Mammendorf, Germany) UV/IR cut filter and a UV-Pass filter in a standardised setting at each field site. We adjusted F-stops, shutter speeds, and ISO to local field conditions. We illuminated each specimen using both ordinary incandescent and UVB light bulbs (Reptizoo; Miami, USA) with standardised angles, placement, and diffusion to minimise specular reflectance, or using natural ambient light when electrical power was unavailable in remote locations (~10% of our data). We photographed specimens against a blue, black, or white matte background (Hengming; Guangzhou, China) with a 40% grey reflectance standard (Labsphere; North Sutton, USA) to standardise photos across variation lighting conditions and a 50 mm scale bar. We performed spectrometry measurements on a subset of snakes from a series of patches representing each unique colour on a given snake (Supplementary Fig. 2) using an Ocean Insight Flame Miniature Spectrometer (Model: FLAME-S-UV-VIS ES) with a PX-2 Pulsed Xenon Light Source and QR400-7-SR Light Source cable (Ocean Insight; Orlando, USA) calibrated using a Labsphere 99% white reflectance standard (Labsphere; USA).

### Ecological characterisation

We used primary literature to characterise habitat and diel activity patterns for snakes. We assigned species as arboreal following Harrington et al.[45], fossorial following Cyriac and Kodandaramaiah[46], or aquatic following Murphy[47], and remaining species as terrestrial with quality control via our field surveys above. We assigned diel activity states as diurnal or nocturnal via Harrington et al.[45] and field guides (Supplementary Data 3). Eight species were not assigned a primary diel activity pattern due to a lack of available information. We assigned sex using a combination of probes and observations of everted hemipenes in males or by palpating developing offspring in females. We determined snake age classes (juvenile, adult) by ranking measured snout-vent lengths (SVL) within species to calculate sex-specific size at maturity thresholds for each species or from published literature[48].

### Manual scoring of UV reflectance from photographs

We quantified UV reflectance for each snake by comparing UV-pass photos using two independent observers (HLC & JDC). First, we scored the dorsal side of each snake for the presence or absence of UV reflectance by comparing pixels on the snake's body to the 40% reflectance standard because pixels without reflectance looked

uniformly dark in comparison (e.g., Fig. 2a, Supplementary Fig. 2), and then we qualitatively estimated the overall proportion of the body with this reflectance (coverage). Because most photographs were illuminated in the same way, we could identify small patches of specular reflectance (mirror-like reflection of light from a shiny scale surface) by their high brightness and repeatable location on the parts of the body in a direct angle toward the light sources (e.g., Supplementary Fig. 1, 6), and we treated those patches as having "no scoreable data". We then recorded the location of the remaining UV reflectance as being on either the head, body, tail or any combination of the three. The corresponding colour in the visible colour range was determined by comparing the UV-pass photo to the visible colour photo. We also qualitatively scored the brightness of the UV-reflective colour as being either less, more or of similar brightness to the 40% colour standard. We then repeated these steps for the ventral surface of each animal. After both scorers had completed the above process, we identified and resolved the few discordances about UV presence or absence via discussion or by an independent third author (ADR). Then, scores were combined into a snake's UV reflectance index by first averaging scores across the two observers, and then multiplying the mean brightness score by the mean proportion of the body with UV reflectance (coverage). We performed these calculations separately for dorsal and ventral surface indices and summed them for an overall reflectance index per snake, and all three of these indices were retained for downstream statistical analysis.

## Quantitative colour pattern analysis and receiver cone catch models

To generate multispectral images (Supplementary Fig. 10), we combined pairs of visible and UV camera RAW images (Nikon NEF files) for the subset of images with the best photographic quality to minimise analytical error. We then normalised across images using the 40% Spectralon reflectance standard included in each photo to create a 5-channel (R, G, B, uvB, and uvR) multispectral image. We excluded background pixels using the corresponding binary mask produced by Batch-Mask[49] and set the scale in ImageJ[50] from a 50 mm scale bar included in each image. UV reflectance was calculated as the sum of uvR and uvB channels across each snake. We note that there was no straightforward way to exclude specular reflectance from quantitative analysis of multispectral images, so much of it was interpreted as UV reflectance in this analysis (see Supplementary Note 1 and Supplementary Fig. 6 for detailed comparison of the manual vs. ImageJ metrics, which were largely congruent). For patch measurements (Fig. 2c), we took the mean of three independent measurements for each channel for each colour on individuals using the polygon and measure tools in ImageJ. To process images for receiver analyses, we used the micaToolbox ImageJ plugin and quantitative colour pattern analysis (QCPA) pipeline[20,51] to produce cone-catch images and run edge intensity analyses. We generated cone-catch models for our camera using the chart-based procedure in micaToolbox with a photographed set of UV-reflective pastels arranged in a grid and measured their precise spectral reflectance curves using a Flame-S-XR1 spectrometer from Ocean Insight. We sourced receptor sensitivities from those available in micaToolbox[20] (bluetit [i.e., UVS bird], peafowl [VS bird], human, gecko [lizard], and dog) and from the literature (snake[21]). Of these, the UVS bird, lizard, and snake have cones with peaks in the UV range (372 and 362 nm), while the dog and peafowl have short-wavelength sensitive cones with peak sensitivities above the UV (430 and 433 nm, respectively) with declining (but non-zero) sensitivity in the UV range. We classified these visual systems as UV-weak in Fig. 4 to distinguish them from the ultraviolet-sensitive (UVS) bluetit, gecko, and snake visual systems, but our visual modelling approach accounts for this non-zero UV sensitivity. All receptor sensitivities ($\lambda_{max}$) are given in Supplementary Table 1. In cases where Weber fractions were not known (gecko and snake), we tested multiple values based on the range of Weber fractions of other visual systems in our dataset and in the literature (e.g., all Weber fractions equal, SWS > LWS). We found that changes in Weber fraction did not qualitatively affect our results (data not shown), so here we report values using Weber fractions of 0.05 for all channels for both gecko and snake.

We then converted the multispectral images to cone-catch images and ran each through the QCPA framework. Briefly, this involved (1) masking out the image background using a region-of-interest (ROI) generated by Batch-Mask[49]; (2) Gaussian acuity correction for a viewing distance of one meter; (3) receptor noise-limited (RNL) filtering and clustering to group colours which would not be distinguishable by that viewer at that distance; and (4) local edge intensity analysis (LEIA) and calculation of other metrics of visual contrast (see Supplementary Fig. 10 and van den Berg et al.[51] for details). To directly assess the perceived chromatic contrast of entire snake colour patterns, we used the mean chromatic colour distance ($\Delta S$, the Euclidean distance between points in receiver colour space) as calculated by the LEIA analysis on the RNL-clustered image. Chromatic $\Delta S$ summarises the degree of colour contrast as a distance metric between two patches (e.g., $\Delta S$ between two pixels of the same colour will be 0) rather than directly comparing average chromaticity, and mean chromatic $\Delta S$ ($\overline{\Delta S}$) represents an average of these values across the whole organism. For example, the blue, red, and yellow plumage of a scarlet macaw would have a higher chromatic $\overline{\Delta S}$ than a uniformly green bird, even if they might have the same average chromaticity. As the goal of this analysis was to compare snake appearances across viewers with differing receptor sensitivities, we kept as many values as possible constant between QCPA runs for different cone-catch models and used framework defaults where appropriate (e.g., viewing distance of one meter, see Supplementary Note 2 and Supplementary Table 1 for all modelling parameters). For cone-catch models that did not include a luminance channel for edge intensity analysis (dog, human, and snake), we created one using the 'Create luminance channel' function by averaging together photoreceptor channels following the literature (Supplementary Table 1).

## Statistical and phylogenetic analysis

To account for phylogenetic uncertainty, we performed all species-level analyses across two different time-calibrated phylogenies with the best coverage across snake species in our dataset[52,53]. We pruned each tree to include only species with molecular data that were also present in our dataset, using intragraneric tip substitution to place congeners where possible (yielding a total of 104 species for analyses with one tree[52] and 95 species for the second tree[53] that had reduced sampling of non-colubrids). To test the effect of ecology (habitat) and diel activity on UV reflectance while controlling for phylogenetic non-independence, we ran phylogenetic linear models[54] and phylogenetic ANOVAs (in 'phytools'[55]) on both manually-scored and computer-scored metrics. We note that the computer-scored metric required the highest quality photos in RAW format, in which the visible and UV photos of the specimen could be perfectly aligned, which was not always possible (e.g., if the specimen was moved slightly in between photos). Usable image pairs represent a subset of ~30% of all snakes measured and a concomitant reduction in statistical power ($N = 65$ species). For intraspecific analyses, we used a series of mixed-effect models to test if sex or age predicts UV reflectance while controlling for clade as a random effect. To avoid singularity issues with our models, we used a reduced data set containing the four clades for which we had enough samples across all age and sex classes (Colubrinae, Dipsadinae, Elapidae, and Viperidae; $N = 410$ individuals). We ran each analysis as linear and multiple regression models to account for distribution non-normality, but all models yielded broadly congruent results. To calculate degrees of freedom and $P$-values, we used the "Satterthwaite approximation" implemented in the 'lmerTest' package[56]. To assess interactions between body regions (e.g., dorsal

and ventral, heads vs. bodies), we used Chi square analyses for both individual data and species means. To test the relative effects of UV sensitivity and chromacy, we fit a multiple regression model predicting mean colour contrast as a function of a number of cones and UV sensitivity ($\lambda_{max}$; see also Supplementary Table 1). We implemented all analyses in R 4.2.1 and assessed significance at $p = 0.05$ with two-tailed tests.

## Reporting summary
Further information on research design is available in the Nature Portfolio Reporting Summary linked to this article.

## Data availability
The "Scopus" database (https://www.scopus.com) was accessed for the UV function literature review. All raw image data, input files, and materials are fully available on the Deep Blue Data repository from the University of Michigan (https://doi.org/10.7302/2ktf-6k49), and all derivative data are in Supplementary Data 1–3. Source data are provided with this paper.

## Code availability
All code and scripts associated with data analyses are located in Deep Blue Data repository from the University of Michigan [https://doi.org/10.7302/2ktf-6k49].

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

## Acknowledgements

We would like to thank all of the scientists and volunteers who helped catch snakes at each of our field sites, including C. Alarcón Rodriguez, A. Basto, S. Boback, J. Buck, A. Canazas Terán, H. Cárdenas, J. Carlos Cusi, Y. Casanca Leon, P. Cerda, M. Cowan, E. Durand Salazar, M. Fernández, S. Goetz, M.R. Grundler, M.C. Grundler, V. Herrera, M. Holding, I. Holmes, O. Huacarpuma Aguilar, E. Iglesias Antonio, J. Larson, E. Lennia, J. Loza, C. Macahuache Díaz, J. Martínez-Fonseca, M. McIntyre, I. Monagan, T. Moore, Z. Muller, D. Nondorf, G. Pandelis, M.R. Parker, D. Rabosky, I. Russell, R. Santa Cruz Farfán, B. Sealey, T. Smiley, N. Tafur Olortegui, E. Taylor, P. Title, S. Van Middlesworth, E. Vargas Laura, R. Villarcorta Díaz, R. von May, E. Westeen, and C. Whitcher. We also thank UMMZ staff members G. Schneider and B. Hess for their assistance with permits and specimen curation and J. Crowe-Riddell, N. Stepanova, and T. West for their consultation on ecological habitat scoring. M.C. Stoddard, J. Troscianko, and C. van den Berg provided advice on photography set-up and colour analyses. J. Megahan provided the *Helicops* drawing for Figs. 2 and S4, and A. Viol developed the plotting script for Fig. 1. We thank M. Holding, G. Kling, D. Rabosky, E. Tibbetts, T. Wittkopp, and all members of the Davis Rabosky lab group for comments and feedback on manuscript drafts. This project was funded by startup funds from the University of Michigan (A.R.D.R.) and the University of Michigan Museum of Zoology Charles F. Walker award (H.L.C.).

## Author contributions

A.R.D.R., H.L.C., J.D.C. and H.I.W. conceptualised and developed the study. All authors contributed equally to methodology, investigation, and visualisation. H.I.W. led the development of macros in ImageJ, and A.R.D.R. led script development in R. Funding was acquired by A.R.D.R. and H.L.C. A.R.D.R. supervised the project. H.L.C., J.D.C. and A.R.D.R. wrote the original draft of the manuscript, and all authors contributed equally to revisions and supplemental information.

## Competing interests

The authors declare no competing interests.
