## [Peer Review File · Nature Communications]

Ecological drivers of ultraviolet colour evolution in snakesREVIEWER COMMENTS

Reviewer #1 (Remarks to the Author):

This study investigates the presence of UV coloration in 110 snake species using ultraviolet photography and spectrophotometry, and assesses potential drivers (predation, sexual selection) for the evolution of UV signalling in these taxa. In the field of visual ecology, it is certainly interesting to know which species reflect UV colouration. This has been documented in multiple other species, including insects, birds and fish. However, it is challenging to provide convincing evidence that selective pressures act on UV coloration without more detailed visual modelling, understanding how these visual signals are displayed against background habitats, and ideally behavioural evidence of how potential signal receivers responding to UV signals. UV reflectance can also be a by-product of the physical properties of pigments or structural colours. While the number of species sampled is impressive, the broad conclusions that the authors make from the title, 'Predators drive the evolution of ultraviolet colouration in snakes' and conclusory sentences, 'Our results show that predator-driven evolution of UV reflectance across snakes not only exists, but also provides a plausible process by which such patterns are maintained over evolutionary time) are not supported by the data.

Major points

Spectral sensitivity curves and visual modelling: More information needs to be provided on the spectral sensitivity curves shown in Figure 3a, as there are some issues that may impact the visual modelling. It states that curves were from micaToolbox (bluetit [i.e., diurnal bird], human, and dog) and from the literature (owl43 and snake44).' However, two species: Nocturnal bird vision and Dog vision state they are UV-, but clearly show the sensitivity curves extending into the UV. Filtering properties of corneas and lenses (e.g. filtering wavelength below 400nm) and oil droplets in burds may be altering the spectral sensitivity curves and need to be considered, otherwise the results from the modelling may be inaccurate. These details should be specified in the methods section and spectral sensitivity curves should be adjusted to show this.

Chromacy: Also, the choice of taxa viewing the snake's colour patterns also need careful consideration as you have a combination of UV+ and UV- sensitive species but also di-, tri- and tetra-chromatic visual systems. Contrast measures are generally going to be higher for those with higher levels of chromacy.

Nocturnal vision: The result stating that, 'Nocturnal snakes also had significantly higher UV reflectance than diurnal snakes ($t = 2.05$; $P = 0.046$)' does not support the hypothesis that predators are driving the amounts of UV reflectance in colour patterns. In nocturnal environments, most of the UV light from ambient conditions has disappeared. Also, in dim light conditions vertebrates switch over from cone-based vision to rod-based vision, which is not UV sensitive.

Ventral vs dorsal: If UV-contrast was driven by predators, then why is UV colouration also displayed at higher levels on ventral surfaces, as predators are unlikely to view the ventral surface during the predation sequence, 'Dorsal UV reflectance was both lower overall and more concentrated into patches along the body than ventral reflectance ($\chi^2 = 40.206$; $P < 0.001$).'

Minor issues

L154-155: 'One fundamental insight from this work is that none of this UV reflectance is predictable from the snake's visible colour pattern alone.' I do not think that most in the field of visual ecology would hypothesis that UV reflectance could be predicted by visible reflectance, so I find this a strange conclusion from the study. Colours are often described by whether they contain a UV component or they don't, for example: blue versus UV-blue, but there has never been any predictions that I have seen that states that by understanding the reflectance in the visible you would be able to know anything about the amount or presence of UV reflectance of the same colour patch.

The authors state L89 that 'This repeated, non-random evolutionary convergence across habitat specialisations strongly supports the conclusion that predators play a critical role in the evolution of UV reflectance across snake species'; however, the study does not investigate the visual characteristics of the background the snakes are viewed against, so it is hard to make inferences from this.

L251: 'Create luminance channel function by averaging together the long- and medium wavelength-sensitive channels, which is the standard in visual ecology'. This needs a citation and it depends on the taxa. For example, fish are thought to process luminance using their double cones, but also use double cones for processing chromatic information. Whereas birds have single cones for colour vision but are thought to only process luminance information through double cones. So is a little more complicated than stated.

L254: there are multiple ways of measuring contrast, so this needs to be explained in more detail here, rather than rely on referencing the QCPA paper or the supplementary information. This doesn't tell you how these measures are calculated, which is important for the results. For example, in Figure 2, it needs to be made clear what the units refer to.

Reviewer #2 (Remarks to the Author):

Dear Authors,

I have very much enjoyed reviewing one of the first papers to study UV colouration in a broad sample of non-avian, non-lepidopteran species. The authors conclude that UV reflectance in some species of snake has an antipredator function. To do so, the paper tests expected relationships between colour form, sex, age, activity time and habitat in order to infer function. This is a useful and valid approach. Results show that UV reflectance is associated with being a juvenile, arboreality and nocturnality, but not with sex, interpreted as consistent with an antipredator function as opposed to a sexual function of being a neutral trait. As with almost all comparative evidence, this evidence is correlational, but that is the way it is – I commend the general approach taken, the quantity and quality of underlying data, as well as many aspects of the methodology and reporting.

For me, one of the most remarkable findings was that UV reflectance was associated with all visible colours (Fig 3) but imperfectly so (i.e. some reds were UV reflective while others weren't). This gives some confidence that UV reflectance is not just a side-effect of selection on non-UV colour. However, this remains my greatest concern for the interpretation that UV colour may have evolved under selection in snakes for an antipredator function, and that this may sometimes be independent of visible colouration. The large majority of white colours are classified as UV reflective and it could be that the chromophores involved in producing white colour are also involved in the production of other colours, for example to make them brighter. It is hard to tell without statistical analysis, but this appears to be suggested by Fig 1C – while UV reflectance does not seem to be correlated with any of the visible colours except white, within a colour category, visible and UV reflectance appear correlated (it would be good to present these analyses). If this is so, it could be that the UV reflectance is a consequence of selection on visible colour. I think discussion of colour production in snakes generally, and specifically of key study species (for example of *Dipsas* and *Micrurus* pictured in Fig 1.) could give more confidence here. What chromophores are responsible for white colour? Do these also contribute to colour in the examples of other UV reflective visible colours?

My second query, which may affect the conclusions drawn, is how UV reflectance relates to structural wear to the epidermis and moulting. We know these are issues affecting UV reflectance in birds (e.g. doi.org/10.1111/j.1095-8312.2002.tb02085.x). The epidermis of juveniles (and arboreal/nocturnal?) snakes could wear faster and/or moult more frequently, and this could contribute to results. Fig. 2a seems to suggest that individuals can vary in their overall UV reflectance (i.e. intra-individual UV reflectance of different colour patches is correlated – another analysis worth doing), which would be consistent with this idea. Additional measurement of individuals at different stages in the moult cycle would be very instructive and give more confidence

in conclusions in my view.

My final substantial concern, which is mentioned in SI Fig 6 but I think needs to be brought to the main methods, is how gloss was considered and handled in analyses. This is related to the previous point – as well as species differences in gloss, glossiness also varies with moult and condition. Glossiness is interesting in that it is a component of the stimulus that a viewer will see looking at the snake, with recent work showing gloss can have antipredator and sexual functions, but it is different to other aspects of colouration in that it will vary with the relative positions of the illuminate and viewer, skin condition, and lighting directionality. As glossiness depends on dermal microstructure it could also plausibly be involved in locomotion, hydrophobicity etc. Clearly some of the UV reflectance in images is due to specular reflections – it sounds like these were avoided in measurement? What is the reasoning for this? Unfortunately I was unable to access the original image files to get a feel for how glossiness varies within and between species in your sample.

Smaller points:

L31-32 “essential for their communication” – ambiguous – rephrase.

L39 There are other functions of colouration – for example anti-parasite colouration, colour used to lure or flush food, colour used for social signalling. Some of these functions make the same prediction as the anti-predator and sexual hypotheses. Could unique predictions for these functions be tested? If not I think this issue needs to be acknowledged.

L64 What does $P > 0.1$ mean here? “all Pagel’s $\lambda < 0.2$; $P > 0.1$ ” -> “all Pagel’s $\lambda < 0.2$ and $P(\neq 0) > 0.1$ ”?

L130 Most natural surfaces (leaves, soil, rocks, bark, water) have low UV reflectivity, at least in the wavelengths most UV sensitive animals are sensitive to. Exceptions include surfaces like snow, light sand, then objects like fruits and flowers. I can’t access the cited study but it seems to focus on much shorter wavelengths than most UV sensitive animals are sensitive to. So I am quite sceptical of the UV camouflage idea.

L132 Could there be a bit more precision in what is meant? A startle display generally involves some behavioural component, which does not seem to align with the animal being asleep: “predator encounters that wake sleeping snakes”?

L195 “Natural ambient light” – was there any further requirement? In very overcast conditions there would be very little UV environmental light available for photography.

L198 As ColorCheckers are not calibrated beyond visible and they do not seem to be used except to provide a scale bar, maybe just streamline and avoid confusion for future readers by saying the images had a scale bar in them?

L216 What were the criteria for presence/absence and % cover? As you have calibrated image data and perform a quantitative analysis on a subset of data, it would be informative to explain why you subjectively quantified presence/absence and %cover for the main analysis (relates to main points 2 and 3).

L220-221 Was this done using pixel values or by eyeballing? It is not clear.

L263 – I’m confused by this – for interspecific analyses the sampling unit is the species – why is there a loss of statistical power? Or did the species sample change?

Fig 2 B&C – wouldn’t it be clearer if the same colour categories were presented in both (i.e. add ‘blue’, ‘orange’)?

Fig 3 b – colour contrast to what?

SI Fig. 7 I think it should be clearer in the main text that results in several instances depended on the phylogeny used, and that quite a few results are borderline significant. In terms of the hypothesis testing the manual and computer scored metrics rarely produce the same result (1/7 significant results are common).

Yours sincerely,

Will Allen

Reviewer #3 (Remarks to the Author):

I commend the authors for their commendable efforts in collecting what is evidently a challenging

dataset, and the results presented in the manuscript are indeed intriguing. However, I have significant reservations regarding both the theoretical foundation and the methodology employed in this paper.

In the introduction, the authors argue that UV-coloration (or UV-reflectance) is often overlooked because it is cryptic to human perception, and its significance is primarily understood in the context of "reproductive biology" (pollination biology and sexual selection). I respectfully disagree with this assertion. The phenomenon of animals perceiving different wavelengths, including those invisible to humans, has been well-established for decades. A simple search on Web of Science yields over a hundred papers that address UV-reflection in various contexts, including predator-prey interactions, its significance in egg-rejection behavior among parasitic birds, parental favoritism, and social interactions. Consequently, it would be more appropriate to acknowledge that while UV-reflection is not ignored, its ecological and evolutionary importance remains an active area of investigation.

The authors make a bold claim that "UV-coloration in snakes is driven by predators," and they support this assertion with several key observations from their data: 1) the reduced UV reflectance on the dorsal side of snakes, 2) the heightened UV reflectance in juveniles, and 3) the increased UV reflectance in arboreal snakes.

In my assessment, deducing such far-reaching conclusions from these observations is not justified. Firstly, there is a lack of evidence establishing a direct link between UV reflectance and predation risk in snakes. In other species, particularly butterflies and moths, the evidence regarding the impact of UV reflectance on predation risk is inconclusive, with some studies suggesting an increase in risk, while others find a decrease or neutrality. The authors seem to assume that UV reflection serves as a warning signal to predators, but the exact mechanism remains unclear (no note about venomousness of snakes for example). Moreover, this assumption raises questions about why juveniles, presumably more vulnerable, exhibit higher reflectance, and why the dorsal side of adult snakes, exposed to predators, displays lower UV reflectance. To address these issues, I strongly recommend that the authors conduct experiments or provide alternative evidence to establish the connection between UV reflection and predation risk in snakes.

Considering that the majority of snakes are nocturnal (with nocturnal species displaying stronger UV-reflection), their visibility to natural predators depends on both natural illumination (irradiance) and the predators' ability to perceive objects in low-light conditions. To better understand the relevance of UV reflection, it is crucial for the authors to calculate absolute quantum cone catch values for the sample patches of interest (preferably also including natural backgrounds) for predators' photoreceptor types. Such calculations require data on natural light levels, combining information on the light source's photonic emissions, the number of reflected photons, and the absorption by photopigments. In essence, this analysis will determine whether the objects of interest, especially UV-reflectance, are visible against their natural backgrounds in low-light conditions to relevant predators.

Regarding snake anatomy, it is worth exploring the possibility that the heightened UV reflection in juveniles and arboreal snakes is non-adaptive or a byproduct of other factors. For instance, juveniles may have thinner skin and less developed scales compared to adults, while arboreal snakes may possess softer skin and less rough scales. If for example the oily liquid between the alpha-layer and beta-layer of snake skin, would be UV-reflective (many oils are), may simply reflect through the thin skin of juveniles? This hypothesis warrants investigation.

In conclusion, while the authors have undertaken commendable work and presented intriguing results, there is a need for a more robust theoretical foundation and a refined methodology to support their claims. Further experiments and analyses are essential to establish a direct link between UV reflection and predation risk in snakes.

Response to Reviewers

Thank you very much for considering our revised submission. The referee comments were very helpful in strengthening the manuscript, especially in clarifying the methods and contextualising the results more carefully. We believe this revised version is much improved over the original. We thank the referees and editorial team for their time and care in giving thoughtful feedback.

Our principal revisions to this manuscript include:

- Revisions to the title, abstract, and throughout the manuscript to better align with uncertainty in causality in the absence of direct experimental evidence on predator perception and response
- New analyses: we have incorporated new receiver visual systems to better test the contributions of chromacy and UV sensitivity to snake colour conspicuousness
- Addition of a new literature review of papers testing the function of UV colouration and a new Figure 1 to support conceptual framing and significance

Referee comments are in regular text and **our responses are bolded**. Please note that for logical flow, we have responded to the reviewers in the reverse order of Referee #3, #2, and then #1.

Referee #3

I commend the authors for their commendable efforts in collecting what is evidently a challenging dataset, and the results presented in the manuscript are indeed intriguing.

Thank you for your thoughtful engagement with the work - yes, snakes in the tropics are very challenging (but exciting) to collect in this breadth and depth!

However, I have significant reservations regarding both the theoretical foundation and the methodology employed in this paper. In the introduction, the authors argue that UV-coloration (or UV-reflectance) is often overlooked because it is cryptic to human perception, and its significance is primarily understood in the context of “reproductive biology” (pollination biology and sexual selection). I respectfully disagree with this assertion. The phenomenon of animals perceiving different wavelengths, including those invisible to humans, has been well-established for decades. A simple search on Web of Science yields over a hundred papers that address UV-reflection in various contexts, including predator-prey interactions, its significance in egg-rejection behavior among parasitic birds, parental favoritism, and social interactions. Consequently, it would be more appropriate to acknowledge that while UV-reflection is not ignored, its ecological and evolutionary importance remains an active area of investigation.

We took this comment very seriously, and we of course agree that UV coloration has been studied in a number of contexts and clades over many years. However, it was unclear to us how uniformly distributed this research attention was overall. If a bias towards reports of UV traits related to a particular function did exist, we needed to quantify its magnitude - or

its absence. So, we performed a large, standardised literature search as suggested, compiling and evaluating a database of 2,400 published articles matching to inclusive search terms about all UV colour across taxa. While many of these studies proved not to be relevant in an ecological context (*e.g.*, UV light in laboratory methodologies), we identified 511 articles across major taxonomic groups that tested for either the a) function or b) perception of UV colouration in nature. We then classified these articles via abstracts and text by the ecological mechanism underlying UV reflection, ultimately binning by major functional classes of reproduction, survival, both, unknown, or other. We have added a new Supplementary Data File with all of these manuscripts and our scores (Supplementary Data 3), included a new figure showing counts in these different categories (now Fig. 1), and revised sections of the Introduction, Conclusions, and extensively in the Methods to reflect these changes.

Briefly, we concluded that for the nearly 300 studies in this database that successfully inferred an ecological function of UV colour (not just its perception), the number reporting functions related to reproduction vastly outnumbered those related to survival. These results are shown in the new main text Fig. 1, in which we displayed raw counts for a) all taxa, and then b) only vertebrates, plotted in each panel by both subclade and function.

However, to facilitate interpretation of these results by our author team, we also generated a null distribution for counts under the assumption that functions related to reproduction and survival were reported equivalently, within and among clades (see Key). We ultimately decided this comparison to a null distribution was too bulky for the manuscript, but we are including the full taxa version below for reviewers to evaluate (see below comparing expected [null distribution] counts to observed counts, numbers also noted in outer arcs, in the same style as Fig. 1). This exercise clearly demonstrates the magnitude of observed skew towards both reproduction and vertebrates/plants relative to what would be expected under a uniform model. Thus, we chose to retain the general conceptual framing and significance that pertained to this reviewer concern. However, we also agree (and note in our revised ms, lines 41–44) that the literature on this topic could easily be a biased subset of the true distribution of functions in nature for a range of reasons, including interests of the research community and relative ease with which certain functions can be studied.

Revised introduction, Lines 41-46: “However, this relative wealth of studies on reproduction-related mechanisms rather than survival could arise not because it is an accurate reflection of the true distribution of function in nature, but simply because species that use colour patterns primarily for crypsis or concealment have been understudied for UV colouration. In theory, multiple types of selection can affect colour evolution, and these processes are hypothesised to produce different signatures on the standing variation of UV colouration within and among species¹⁰.”

The authors make a bold claim that "UV-coloration in snakes is driven by predators," and they support this assertion with several key observations from their data: 1) the reduced UV reflectance on the dorsal side of snakes, 2) the heightened UV reflectance in juveniles, and 3) the increased UV reflectance in arboreal snakes. In my assessment, deducing such far-reaching conclusions from these observations is not justified. Firstly, there is a lack of evidence establishing a direct link between UV reflectance and predation risk in snakes. In other species, particularly butterflies and moths, the evidence regarding the impact of UV reflectance on predation risk is inconclusive, with some studies suggesting an increase in risk, while others find a decrease or neutrality. The authors seem to assume that UV reflection serves as a warning signal to predators, but the exact mechanism remains unclear (no note about venomousness of snakes for example). Moreover, this assumption raises questions about why juveniles, presumably more vulnerable, exhibit higher reflectance, and why the dorsal side of adult snakes, exposed to predators, displays lower UV reflectance. To address these issues, I strongly recommend that the authors conduct experiments or provide alternative evidence to establish the connection between UV reflection and predation risk in snakes.

Revised as suggested; we have removed “predators” from the title, as well as extensively revised language in the abstract, introduction, results, and conclusions to better align with uncertainty in causality. We have also provided better logical framing in the Discussion about how our results fit together within the possible mechanisms that could generate survival benefits to UV colouration. We agree that behavioural experiments on both predator and snake responses are the next round of experiments and data collection necessary to fully test these hypotheses - now informed by this new knowledge of which parts of a snake’s body are relevant to this test, and hopefully inspiring working groups beyond our own as well.

Additionally, we agree fully with the reviewer that any assumption that UV primarily serves as warning coloration would be premature and, in our view, less plausible than several alternative hypotheses (Lines 157-163). In our revised manuscript, we better define these potential mechanisms as either 1) crypsis, or 2) a “startle” or “signal boosting” effect that visible colour markings serve in defensive displays performed across the vast majority of snakes that are not dangerously venomous to predators (*e.g.*, of which one example is the amazing display of *Dasypeltis*, which has no teeth at all: <https://www.youtube.com/shorts/eXJ2tBxaOUE>; see also Gans & Richmond 1957, *Copeia*). We also note that like in this

***Dasypeltis*, these mechanisms are not mutually exclusive (e.g., blotching serves a crypsis function when stationary, signal boost when in motion).**

Extensively revised discussion, Lines 153-206.

Considering that the majority of snakes are nocturnal (with nocturnal species displaying stronger UV-reflection), their visibility to natural predators depends on both natural illumination (irradiance) and the predators' ability to perceive objects in low-light conditions. To better understand the relevance of UV reflection, it is crucial for the authors to calculate absolute quantum cone catch values for the sample patches of interest (preferably also including natural backgrounds) for predators' photoreceptor types. Such calculations require data on natural light levels, combining information on the light source's photonic emissions, the number of reflected photons, and the absorption by photopigments. In essence, this analysis will determine whether the objects of interest, especially UV-reflectance, are visible against their natural backgrounds in low-light conditions to relevant predators.

While about half of the species in our dataset are nocturnal, the arboreal species in particular are still visible during the day while they sleep in trees. In that sense, the ability of a nocturnal snake to remain inconspicuous during daytime hours may be at least as important as its ability to do so at night. Furthermore, many nocturnal snakes begin moving early enough in the late afternoon that they may draw the attention of diurnal visually-oriented predators (revised manuscript, Lines 164-173). However, we broadly agree with this comment that an analysis of quantum cone catch values under nocturnal viewing conditions has the potential to be extremely informative and to provide qualitatively different information than the analysis we do present. These, and extensive natural background data, would be very exciting data to collect in the future.

However, we elected not to compute absolute quantum cone catch values for patches of interest on the snakes in our images because we believe that these values would be difficult to interpret without first gathering extensive data on background (= ambient environmental) UV reflectance. We chose to focus on measures of internal contrast because they have ecological relevance but do not require comparisons with the background and would allow us to photograph and analyse each snake in a standardised way. In essence, the purpose of our visual modelling analysis was to ask whether the observed patterns of UV reflectance have the *potential* to affect how receivers view them. We chose a daylight illuminant primarily because UV light is more abundant during the day than at night, meaning selection pressure may be greatest for UV reflectance under daylight conditions.

Regarding snake anatomy, it is worth exploring the possibility that the heightened UV reflection in juveniles and arboreal snakes is non-adaptive or a byproduct of other factors. For instance, juveniles may have thinner skin and less developed scales compared to adults, while arboreal snakes may possess softer skin and less rough scales. If for example the oily liquid between the

alpha-layer and beta-layer of snake skin, would be UV-reflective (many oils are), may simply reflect through the thin skin of juveniles? This hypothesis warrants investigation.

Revised as suggested: We have now added an extensive supplementary discussion section (Supplementary Text) on 1) skin, 2) scales, 3) shed/moult cycles, and 4) shine to address these concerns (and those of Reviewer #2).

Importantly, we found that ventral scales – the biggest, thickest scales on all snake ecotypes and age classes – tended to have the highest UV reflectance, thus suggesting that softer or thinner scales are unlikely to explain either higher reflectance in juveniles or in arboreal taxa. This point is discussed more extensively in our revised Supplementary Text, as well as how scale property considerations may impact the study of every colour in reptile scales, rather than being something specific to UV wavelengths. We also acknowledge there that a better understanding of the cellular mechanisms underlying UV colour, including oils, will be important for resolving when and how scale properties themselves may be contributing to variation in UV reflectance. We agree fully that this warrants further investigation.

In conclusion, while the authors have undertaken commendable work and presented intriguing results, there is a need for a more robust theoretical foundation and a refined methodology to support their claims. Further experiments and analyses are essential to establish a direct link between UV reflection and predation risk in snakes.

Thank you again for your helpful comments!

Referee #2

I have very much enjoyed reviewing one of the first papers to study UV colouration in a broad sample of non-avian, non-lepidopteran species. The authors conclude that UV reflectance in some species of snake has an antipredator function. To do so, the paper tests expected relationships between colour form, sex, age, activity time and habitat in order to infer function. This is a useful and valid approach. Results show that UV reflectance is associated with being a juvenile, arboreality and nocturnality, but not with sex, interpreted as consistent with an antipredator function as opposed to a sexual function of being a neutral trait. As with almost all comparative evidence, this evidence is correlational, but that is the way it is – I commend the general approach taken, the quantity and quality of underlying data, as well as many aspects of the methodology and reporting.

Thank you!

For me, one of the most remarkable findings was that UV reflectance was associated with all visible colours (Fig 3 [author note: original Fig. 1c?]) but imperfectly so (i.e. some reds were UV reflective while others weren't). This gives some confidence that UV reflectance is not just a side-effect of selection on non-UV colour. However, this remains my greatest concern for the

interpretation that UV colour may have evolved under selection in snakes for an antipredator function, and that this may sometimes be independent of visible colouration. The large majority of white colours are classified as UV reflective and it could be that the chromophores involved in producing white colour are also involved in the production of other colours, for example to make them brighter. It is hard to tell without statistical analysis, but this appears to be suggested by Fig 1C – while UV reflectance does not seem to be correlated with any of the visible colours except white, within a colour category, visible and UV reflectance appear correlated (it would be good to present these analyses). If this is so, it could be that the UV reflectance is a consequence of selection on visible colour. I think discussion of colour production in snakes generally, and specifically of key study species (for example of *Dipsas* and *Micrurus* pictured in Fig 1.) could give more confidence here. What chromophores are responsible for white colour? Do these also contribute to colour in the examples of other UV reflective visible colours?

Revised as suggested regarding what is known about chromatophores, pigments, and structural colours, as we have now added a supplementary discussion point about “white” (Supplementary Text, paragraph four): “The third concern is about whether the cellular mechanisms that could be producing UV reflectance may be correlated with a visible colour that itself is under selection, which would suggest our results could be a by-product of selection in the visible spectrum. We did measure white colours as UV reflective more often than any other visible colour (Fig. 2c, Fig. S4b; note that what humans perceive as “white” generally represents high reflectance across all visible wavelengths, see Fig. S2), but we do not yet have enough cellular data in snakes to assign structural or pigment-based mechanisms to most colours other than those produced by melanins (blacks) or pteridines (reds; with minimal to no carotenoids¹), including UV or white. In lizards, it appears at least some UV reflectance is caused by structural colours rather than by pigments⁷, and this mechanism of a structural UV colour overlaying pigments of any variety would be consistent with our finding that many visible colours can have UV reflectance, including black (Fig. S1c; Fig. S4b; Fig 2c). While we cannot currently perform a formal test of this hypothesis, we consider the cellular mechanisms underlying UV colour to be the next big frontier in this field necessary to resolve some of these outstanding questions.”

This second suggestion - a formal analysis of whether UV reflectance correlates among and/or within colours - was a very intriguing one that we thought a lot about how to implement appropriately in the context of biological rather than just statistical significance. We chose a Chi-squared approach to test if observed occurrences of colours that are also UV reflective deviated from expected values under a random distribution (now a new Fig. S4b). Not surprisingly, UV reflective white patches occurred at a significantly higher frequency than what would be expected under the null model. However, it is worth noting that every visible colour is reported to have at least one occurrence of UV reflectance and many colours (*i.e.*, yellow, black, green, and brown) actually reflected significantly lower amounts of UV than expected, suggesting that selection for bright colouration alone does not drive UV reflectance. In relation to actual histology of the snake scales, it is possible that melanins and other non-white colour producing pigments may suppress the occurrence of

UV reflectance, especially on the dorsal surfaces of snakes (the surface most exposed to predators). But overall, this analysis revealed that many colours, not just white, are correlated with UV production, likely favouring a structural colour mechanism overlaying pigments of any variety.

My second query, which may affect the conclusions drawn, is how UV reflectance relates to structural wear to the epidermis and moulting. We know these are issues affecting UV reflectance in birds (e.g. doi.org/10.1111/j.1095-8312.2002.tb02085.x). The epidermis of juveniles (and arboreal/nocturnal?) snakes could wear faster and/or moult more frequently, and this could contribute to results. Fig. 2a seems to suggest that individuals can vary in their overall UV reflectance (i.e. intra-individual UV reflectance of different colour patches is correlated – another analysis worth doing), which would be consistent with this idea. Additional measurement of individuals at different stages in the moult cycle would be very instructive and give more confidence in conclusions in my view.

Revised as suggested; We have now added an extensive supplementary discussion section (Supplementary Text) on 1) skin, 2) scales, 3) shed/moult cycles, and 4) shine to address these concerns (and those of Reviewer #3).

Importantly, we note that each individual can also be used as a control for itself in assessing the impact of shed state. Different scales on the same body are all at the same point in the shed cycle due to snakes moulting their entire epidermis as a single layer, and these scales can be directly compared. For example, the ventral scales of the “unreflective” snake on the left in original Fig. 2a (now Fig. 3a) are actually quite strongly UV reflective, even though the ventral scales are at the same shed cycle point as this snake’s dorsal scales. We include below another image of this same individual that emphasizes this comparison by simultaneously showing some ventral scales and some dorsal scales (some ventral scales are also visible in the original figure). This result was also true more broadly across the dataset, as we reported in the original manuscript that “UV reflectance on one body surface did not reliably predict reflectance on the other surface (all $R^2 = 0-0.11$)” (Lines 76-77). We thus concluded that our overall results are unlikely to reflect something as simple as shed state.

Additionally, while our efforts were not robust enough to include even in the supplement, we did trial one round of exploration on the dorsal UV reflectance of four adult captive snakes with known shed cycle information to ensure that they weren't immediately post-shed, nor imminently preparing to shed - in other words, rather than first testing repeated measures on the same snake to test whether it had the same UV reflectance at multiple time points, instead testing multiple snakes all at similar shed cycle points to ensure they all had different UV reflectance. All of these snakes were similar visible shades of brown, black, and white/cream (below).

However, it is difficult to keep live snakes still enough for the required slow shutter speeds without significant unwanted medical intervention with veterinary oversight (these snakes were not intended for museum preservation like the others). Thus, our pilot efforts yielded unusable photos with motion blur, at non-standard illuminance angles, and without colour standards (see below), so we did not pursue repeated measures. However, these informal observations were fully consistent with Fig. 3's display of variation across the phylogeny that variation among species was significant - and the expectation that no amount of shedding seems likely to make the *Lichanura* (below, panel d) UV reflective, as all that is visible on both dorsal and ventral surfaces is specular reflectance (and this exercise reminded us how amazed we are at having generated a dataset with so little specular reflectance visible to interfere with analysis - it's a difficult lighting challenge). Note that none of these four species were in the original dataset, and we've not included them further.

My final substantial concern, which is mentioned in SI Fig 6 but I think needs to be brought to the main methods, is how gloss was considered and handled in analyses. This is related to the previous point – as well as species differences in gloss, glossiness also varies with moult and condition. Glossiness is interesting in that it is a component of the stimulus that a viewer will see looking at the snake, with recent work showing gloss can have antipredator and sexual functions, but it is different to other aspects of colouration in that it will vary with the relative positions of the illuminate and viewer, skin condition, and lighting directionality. As glossiness depends on dermal microstructure it could also plausibly be involved in locomotion, hydrophobicity etc. Clearly some of the UV reflectance in images is due to specular reflections – it sounds like these

were avoided in measurement? What is the reasoning for this? Unfortunately I was unable to access the original image files to get a feel for how glossiness varies within and between species in your sample.

Revised as suggested; we have now added a more extensive supplementary discussion section (Supplementary Text) on shine and also revised the main text methods to explain how human scorers avoided it (Lines 312-316) but computer scoring included it (Lines 336-339).

As in the collage above, specular reflectance was indeed the most annoying of the nuisance variables in our dataset, which as this reviewer notes is why we highlighted it in Fig S6. We did standardise the direction, position, and diffusion of all lights in order to both minimise and standardise where specular reflection appeared (Lines 284-287). This standardisation of directionality allowed human observers to simply avoid those patches as having “no scoreable data”, while the computer scores did include specular reflectance as UV reflectance. One test of whether this effect drove our ecological results was to directly compare the two metrics - one that fully included gloss and one that fully avoided it - and their persistent positive correlation (Fig S6b-d) suggested that the overall measurement impact of glossiness wasn't the issue of greatest concern for the global dataset (although it did explain much residual variation specifically for the snakes with very low UV indices under manual scoring, low x-axis values in Fig S6b,c).

However, we are also concerned that this reviewer was unable to access the entire dataset of photos that we had made available for review - please see the link below and let us know if you have any other difficulties! We invite reviewers to flip through them as desired. https://drive.google.com/drive/folders/1EvFojHAqmO13hr1yY1U0AcogM_7KxbyO (formal deposition embargoed until acceptance at: <https://doi.org/10.7302/2k49>)

Smaller points:

L31-32 “essential for their communication” – ambiguous – rephrase.

Revised as suggested, Lines 30-32, “However, the ability to perceive ultraviolet (UV) wavelengths (300-400 nm) is found in many other animals, and UV colour is used frequently in visual communication within and across species^{3, 4}.”

L39 There are other functions of colouration – for example anti-parasite colouration, colour used to lure or flush food, colour used for social signalling. Some of these functions make the same prediction as the anti-predator and sexual hypotheses. Could unique predictions for these functions be tested? If not, I think this issue needs to be acknowledged.

Revised to clarify, although much of this revision as it pertains to non-reproductive hypotheses is actually now in the Discussion (Lines 153-186). We appreciate that some juvenile snakes do use brightly coloured caudal lures to attract prey, which would result in

similar expectations regarding ontogenetic colour shifts, but would be restricted to tails, not heads/chins as we found (Line 79). However, caudal lures are documented mainly in a few pit vipers and in death adders (Elapids from Australia), and these known lurers are represented by only five pit viper species in our dataset. Additionally, our extensive literature review above also suggested that most tests of intraspecific communication couldn't rule out signals relevant for reproduction (mate choice, access to breeding sites, honest signals related to immune fitness for reproduction, etc.), although we scored some of these studies as "both" if survival was explicitly addressed, but mainly these would conflate with the hypotheses for reproduction and survival. Because we didn't find outcomes supporting sexual or social signalling in our own study, we focused this clarification on mechanisms underlying survival and their lack of mutually exclusive predictions without additional data types.

L64 What does $P > 0.1$ mean here? "all Pagel's $\lambda < 0.2$; $P > 0.1$ " -> "all Pagel's $\lambda < 0.2$ and $P(\neq 0) > 0.1$ "?

Revised as suggested, Lines 74-76: "UV reflectance on both dorsal and ventral surfaces showed little phylogenetic signal, even when accounting for phylogenetic uncertainty (all Pagel's $\lambda < 0.2$, and all $p > 0.1$)."

L130 Most natural surfaces (leaves, soil, rocks, bark, water) have low UV reflectivity, at least in the wavelengths most UV sensitive animals are sensitive to. Exceptions include surfaces like snow, light sand, then objects like fruits and flowers. I can't access the cited study but it seems to focus on much shorter wavelengths than most UV sensitive animals are sensitive to. So I am quite sceptical of the UV camouflage idea.

Partially revised as suggested - testing background reflectance is the next critical step that requires additional data, which we now highlight better in the Discussion (Lines 203-207). However, we note that trees in the tropics are literally covered in other life - fungi and lichens, epiphytes of all varieties, detritus, etc. - that could generate highly complex background patterns of UV reflectance and absorbance. We simply don't have spectral reflectance profiles for the vast majority of these habitat elements across the tropics, as most work on reflectance spectra of plants and lichens is done at high latitude and/or high elevation. However, we did find (and now additionally cite) one study that measured both Neotropical lichens and their host tree bark in Costa Rica and released their raw data. We plotted their data here, and UV reflectance (below 390nm) is common and high in both:

While this is only one paper with one method for measuring reflectance (spectrometer only), we are cautiously encouraged by the plot above and hope our work's publication inspires more research groups to take on this challenge of broadly quantifying background reflectance with both spectrometers and full spectrum imaging. We also revised the Discussion to further clarify the remaining hypotheses and what would be needed to test them, especially as pertains to crypsis, which remains the simplest hypothesis and therefore first in line to reject.

Revised Discussion, Lines 165-173: “First, UV colouration could function to reduce detection by diurnal predators while nocturnal snakes sleep during the daytime, as both plants³⁰ and their epiphytes³¹ can have UV reflectance. In this case, UV patterning could make a snake harder to detect by UV-sensitive birds living in trees, as in peppered moths on crustose lichens³². Nearly all arboreal snakes, both diurnal and nocturnal, are countershaded in the visible spectrum, offsetting lighter bellies with darker dorsal patterning. While some elements of these human-visible patterns are likely neutral, both green patterns (*e.g.*, Fig. S1e) and blotchy patterns (*e.g.*, Fig. 2a) are so universal and repeatable in arboreal taxa that “avoidance of detection” while exposed on tree branches is widely assumed to underlie the maintenance of such colour patterns³¹. Thus, the additional UV reflectance that we have documented to these otherwise broadly-visible pattern elements should not be assumed *a priori* to function differently.”

Ref 31, source of Costa Rican lichen data: Guzmán Q JA, Laakso K, López-Rodríguez JC, Rivard B, Sánchez-Azofeifa GA. Using visible-near-infrared spectroscopy to classify lichens at a Neotropical Dry Forest. *Ecol Indic* 111, 105999 (2020). Raw data: <https://doi.org/10.7910/DVN/Y1J0UO>

Revised Discussion, Lines 203-207: “Experimental tests involving reflectance measurements across the complex background conditions found in nature (*e.g.*, with native lichens, fungi, etc. while in situ on tree branches), combined with predator responses to many combinations of snake colour patterns across these backgrounds, are the critical next step for determining the relative importance of these mechanisms across snake species.”

L132 Could there be a bit more precision in what is meant? A startle display generally involves some behavioural component, which does not seem to align with the animal being asleep: “predator encounters that wake sleeping snakes”?

Revised as suggested, Lines 173-176, “However, UV patterns could additionally function as a sort of startle or “signal boosting” colouration displayed by abruptly-awoken snakes to aid in escape immediately following predator encounters of sleeping snakes in a well-lit daytime environment^{32, 33, 34}.”

L195 “Natural ambient light” – was there any further requirement? In very overcast conditions there would be very little UV environmental light available for photography.

No, there was no further requirement because the only times we used ambient light were as a last resort in some field sites, when logistically we had no access to any electrical power to run the lights (which occurred for ~10% of our data, revised Line 287). These were difficult decisions in which we weighed the benefit of generating at least some quantitative colour data on a species we may never encounter again (each individual snake takes on average ~10-20 person-hours to find in these sites) against the cost of not generating perfectly standardised data. In those cases, we did adjust shutter speeds to accommodate some variation in light levels, and we also used the 40% reflective grey colour standard in each photo to further calibrate across lighting variation (Lines 288-290). It was not perfect, but typically, the ambient light photos primarily suffered from UV *overexposure* rather than *underexposure*.

L198 As ColorCheckers are not calibrated beyond visible and they do not seem to be used except to provide a scale bar, maybe just streamline and avoid confusion for future readers by saying the images had a scale bar in them?

Revised as suggested, Lines 287-290, “We photographed specimens against a blue, black, or white matte background (Hengming; Guangzhou, China) with a 40% grey reflectance standard (Labsphere; North Sutton, USA) to standardise photos across variation lighting conditions and a 50mm scale bar.”

L216 What were the criteria for presence/absence and % cover? As you have calibrated image data and perform a quantitative analysis on a subset of data, it would be informative to explain why you subjectively quantified presence/absence and %cover for the main analysis (relates to main points 2 and 3).

Revised as suggested, clarified methods on Lines 308-312: “First, we scored the dorsal side of each snake for the presence or absence of UV reflectance by comparing pixels on the snake’s body to the 40% reflectance standard because pixels without reflectance looked uniformly dark in comparison (see Fig. 2a), and then we qualitatively estimated the overall proportion of the body with this reflectance (coverage).”

Fundamentally, these choices were simply logistical: some snakes moved/were moved between photos so that mspecks did not align properly, didn’t lie perfectly flat during photography due to body shape (*i.e.*, had a lot of ventral surface visible in a dorsal image or vice versa), or otherwise did not have suitable RAW files. However, all photos were suitable for manual scoring by humans that could easily account for all of this variation. It ended up being quite interesting to compare where the two approaches differed, as they each had different strengths despite being in global alignment (Fig. S6). But overall, the manual scoring of UV reflectance was pretty straightforward and repeatable among scorers (see Data File 1), and we invite the reviewers to compare visible-to-UV “photo pairs” and scores themselves, too. We note that one of our author team members is not a herpetologist but

was able to understand and execute a trial set of independent scores with minimal direction.

L220-221 Was this done using pixel values or by eyeballing? It is not clear.

Revised as suggested, Lines 318-320: “We also qualitatively scored the brightness of the UV-reflective colour as being either less, more, or of similar brightness to the 40% colour standard.”

L263 – I’m confused by this – for interspecific analyses the sampling unit is the species – why is there a loss of statistical power? Or did the species sample change?

Yes, due to the comments two points above, our number of species was lower for the computer-scored analyses, so the species sample changed and thus was reported in this line.

Fig 2 B&C – wouldn’t it be clearer if the same colour categories were presented in both (i.e. add ‘blue’, ‘orange’)?

We assume this comment referred to “Fig 1b,c” as we had no blue or orange in Fig 2. For Fig. 1c (now Fig 2c in the revised manuscript), the colour patches needed to be of a size and location that was reliable to isolate and measure (e.g., bright blue “cheek” stripes in *Leptophis* are visible in both dorsal and ventral photographs during manual scoring by a human, but neither camera perspective is ideal for measuring the lateral side of snake heads). These were the five colour groups in which we were the most confident and had the most robust sampling.

Fig 3 b – colour contrast to what?

Revised as suggested for clarity (now Fig. 4b), and also Lines 366-372: “To directly assess the perceived chromatic contrast of entire snake colour patterns, we used the mean chromatic colour distance (ΔS , the Euclidean distance between points in receiver colour space) as calculated by the LEIA analysis on the RNL-clustered image. ΔS summarizes the degree of colour contrast as a distance metric between two patches (e.g., ΔS between two pixels of the same colour will be 0) rather than directly comparing average brightness, and mean ΔS represents an average of these values across the whole organism. For example, a zebra would have a higher mean ΔS than a grey horse, even if they might have the same average luminance.”

SI Fig. 7 I think it should be clearer in the main text that results in several instances depended on the phylogeny used, and that quite a few results are borderline significant. In terms of the hypothesis testing the manual and computer scored metrics rarely produce the same result (1/7 significant results are common).

These two trees are difficult to directly compare because they have different complements and numbers of species that sample differently across ecological transitions, rather than being a clear test of phylogenetic uncertainty across identical species sets. However, they are the best two (independently derived, importantly) trees available at this time with the best coverage of Neotropical species, and we made this figure because we wanted to be fully transparent about these impacts.

Overall, though, we were pleasantly surprised at how similarly these models and metrics performed, and we think this figure demonstrates that the results were in global alignment rather than discordance. We recommend assessing p-values specifically in the context of the boxplots, as this is the logic for how we evaluated model performance. First, we compared the boxplots between the two *metrics* and asked: were values consistent, such that those that were low in one metric also generally low in another and vice versa, even despite the differences in sample size (answer: yes). Then, given these boxplot distributions of the data by ecological category, were the analyses using the two *phylogenies* in strong conflict: specifically, did they 1) call categories significantly different when the boxplots were actually highly similar (answer: no - although this can legitimately occur with particular trait distributions across a phylogeny, it didn't here), or did they 2) give p-values near 1 for one tree but near 0 for the other and thus show high conflict (answer: no).

Because the Zaher tree was the more underpowered test because it dropped so many of the non-colubrids (which non-randomly represent the earlier diverging taxa on longer branches) that were present in the Tonini tree, we explicitly expected that near-significant but above threshold p-values were likely for tests that were significant with the Tonini tree. This is indeed both the pattern and direction explaining the mild level of discordance between the two tests. We thus upweighted the Tonini results in the main text as the more robust test overall, but presented both in the supplement for the more specialised subset of readers like this reviewer who may be interested in details of model performance across phylogenies.

Yours sincerely,
Will Allen

Thank you for your constructive comments!

Referee #1:

This study investigates the presence of UV coloration in 110 snake species using ultraviolet photography and spectrophotometry, and assesses potential drivers (predation, sexual selection) for the evolution of UV signalling in these taxa. In the field of visual ecology, it is certainly interesting to know which species reflect UV colouration. This has been documented in multiple other species, including insects, birds and fish. However, it is challenging to provide convincing

evidence that selective pressures act on UV coloration without more detailed visual modelling, understanding how these visual signals are displayed against background habitats, and ideally behavioural evidence of how potential signal receivers respond to UV signals. UV reflectance can also be a by-product of the physical properties of pigments or structural colours. While the number of species sampled is impressive, the broad conclusions that the authors make from the title, ‘Predators drive the evolution of ultraviolet colouration in snakes’ and conclusory sentences, ‘Our results show that predator-driven evolution of UV reflectance across snakes not only exists, but also provides a plausible process by which such patterns are maintained over evolutionary time’ are not supported by the data.

Revised as suggested; we have removed “predators” from the title, as well as extensively revised language in the abstract, results, and discussion to better align with the level of uncertainty about causal inference that exists without the important, direct experimental evidence that this reviewer recommends.

Major points

Spectral sensitivity curves and visual modelling: More information needs to be provided on the spectral sensitivity curves shown in Figure 3a, as there are some issues that may impact the visual modelling. It states that curves were from micaToolbox (bluetit [i.e., diurnal bird], human, and dog) and from the literature (owl43 and snake44).’ However, two species: Nocturnal bird vision and Dog vision state they are UV-, but clearly show the sensitivity curves extending into the UV. Filtering properties of corneas and lenses (e.g. filtering wavelength below 400nm) and oil droplets in birds may be altering the spectral sensitivity curves and need to be considered, otherwise the results from the modelling may be inaccurate. These details should be specified in the methods section and spectral sensitivity curves should be adjusted to show this.

Revised as suggested; We have added more detail on the spectral sensitivity curves (new Supplementary Text, Supplementary Table 1), and species with low sensitivity in the UV part of the spectrum are now labelled as “UV-weak” rather than “UV-” in the figure (now Fig. 4a). Note also that in part due to this lack of detail on owl oil droplets, we have exchanged the owl model for the better established violet-sensitive peafowl (curves available in micaToolbox), as both this system and bluetit are both already adjusted for oil droplet filtering.

Chromacy: Also, the choice of taxa viewing the snake’s colour patterns also need careful consideration as you have a combination of UV+ and UV- sensitive species but also di-, tri- and tetra-chromatic visual systems. Contrast measures are generally going to be higher for those with higher levels of chromacy.

Revised as suggested; We have now implemented a much improved order and grouping approach in Fig. 4 (originally Fig. 3), adding not only the peafowl but also gecko visual systems (both well characterised and available in micaToolbox) to create an intuitive

“experimental design” of two systems per chromacy level that vary in UV sensitivity. We also conducted a multiple regression to test the significance of these two factors (chromacy and UV sensitivity) in predicting mean dorsal colour contrast, and yes, both are highly significant.

Revised Results, Line 121-126: “To test whether the higher colour contrast for UV-sensitive visual systems was only the result of these visual systems’ higher chromacy, we fit a multiple regression model predicting colour contrast as a function of chromacy (2, 3, or 4 cones) and UV sensitivity (λ_{\max} for the shortest peak sensitivity of any cone). Both UV sensitivity and chromacy were significant predictors of colour contrast ($P < 0.001$), suggesting that the importance of UV reflectance in our results is not solely a consequence of our choice of visual systems.”

Nocturnal vision: The result stating that, ‘Nocturnal snakes also had significantly higher UV reflectance than diurnal snakes ($t = 2.05$; $P = 0.046$)’ does not support the hypothesis that predators are driving the amounts of UV reflectance in colour patterns. In nocturnal environments, most of the UV light from ambient conditions has disappeared. Also, in dim light conditions vertebrates switch over from cone-based vision to rod-based vision, which is not UV sensitive.

Revised extensively (Lines 153-206) as suggested to clarify that we are considering interactions between predators and arboreal, nocturnal snakes explicitly during daylight hours when they are sleeping on branches. See also the response to Reviewer #2 above.

Ventral vs dorsal: If UV-contrast was driven by predators, then why is UV colouration also displayed at higher levels on ventral surfaces, as predators are unlikely to view the ventral surface during the predation sequence, ‘Dorsal UV reflectance was both lower overall and more concentrated into patches along the body than ventral reflectance ($\chi^2 = 40.206$; $P < 0.001$).’

Revised Discussion to clarify the non-mutually exclusive selection mechanisms (and the direction of selection) that could generate discordance between ventral and dorsal surfaces.

Revised Lines 187-206, “Additionally, our data cannot confirm that selection on UV colouration is directional towards higher UV reflectance, whether broadly on ventral surfaces or on patched dorsal surfaces on nocturnal, arboreal snakes. An alternative explanation is that dorsal UV reflectance is selected against in diurnal and terrestrial snakes, such that UV reflectance is only retained when it is selectively neutral, as on a rarely-visible surface like a belly. We note that these mechanisms are also not mutually exclusive, and both processes could interact simultaneously to produce a snake’s full colour phenotype. However, the latter mechanism would be a wholesale departure from the way UV colouration is currently conceptualised in both animal and plant systems, and it would require further experimental testing. Overall, the pervasive restriction of UV reflectance to patches or distinct body regions on the dorsum suggests that UV reflectance on low-

visibility ventral surfaces may simply be under weaker selection (as in moths³⁵) and that selection on the dorsum is not only strong, but in the direction of signal stabilisation rather than diversification (as in tropical birds³⁸). This result would run counter to most theory on the evolution of “hidden channel” communication, in which UV signals are expected to have reliable species-specific information that should generate increased signal diversity⁴, but would be entirely consistent with crypsis. Experimental tests involving reflectance measurements across the complex background conditions found in nature (*e.g.*, with native lichens, fungi, etc. while in situ on tree branches), combined with predator responses to many combinations of snake colour patterns across these backgrounds, are the critical next step for determining the relative importance of these mechanisms across snake species.”

Also, Revised Lines 166-179: “ Nearly all arboreal snakes, both diurnal and nocturnal, are countershaded in the visible spectrum, offsetting lighter bellies with darker dorsal patterning. While some elements of these human-visible patterns are likely neutral, both green patterns (*e.g.*, Fig. S1e) and blotchy patterns (*e.g.*, Fig. 2a) are so universal and repeatable in arboreal taxa that “avoidance of detection” while exposed on tree branches is widely assumed to underlie the maintenance of such colour patterns³¹. The additional UV reflectance that we have documented to these otherwise broadly-visible colour patterns should not be assumed *a priori* to function differently. However, UV patterns could additionally function as a sort of startle or “signal boosting” colouration displayed by abruptly-awoken snakes to aid in escape immediately following predator encounters of sleeping snakes in a well-lit daytime environment^{32, 33, 34}. High UV reflectance has been found in other nocturnal lineages, especially within insect groups like moths, which also have higher reflectance on hindwings that are hidden at rest but deployed during diurnal disturbance³⁵ in the same manner as snake venters¹¹.”

Minor issues

L154-155: ‘One fundamental insight from this work is that none of this UV reflectance is predictable from the snake’s visible colour pattern alone.’ I do not think that most in the field of visual ecology would hypothesize that UV reflectance could be predicted by visible reflectance, so I find this a strange conclusion from the study. Colours are often described by whether they contain a UV component or they don’t, for example: blue versus UV-blue, but there has never been any predictions that I have seen that states that by understanding the reflectance in the visible you would be able to know anything about the amount or presence of UV reflectance of the same colour patch.

Revised as suggested to clarify intent, as we fully agree with this great example about blue vs. UV-blue, etc. We more meant that researchers should not simply make decisions about which species are likely to have (or not have) UV reflectance because they are making an unconscious judgement about taxa based on their visible patterning, rather than implying that researchers have been assuming that every organism that was a certain colour should additionally have (or not have) UV reflectance.

Revised Discussion, Lines 207-209: “One fundamental insight from this work is that the conspicuousness of an organism’s visible colour pattern should not be the primary motivator for choosing which species are studied for UV colouration.”

The authors state L89 that ‘This repeated, non-random evolutionary convergence across habitat specialisations strongly supports the conclusion that predators play a critical role in the evolution of UV reflectance across snake species’; however, the study does not investigate the visual characteristics of the background the snakes are viewed against, so it is hard to make inferences from this.

Revised as suggested to clarify that these results more strongly reflect a rejection of neutrality and remaining more agnostic about agents of selection, Lines 103-106: “This repeated evolutionary convergence across the same habitat specialisations is inconsistent with a conclusion that neutral processes are driving the evolution of UV reflectance across these snake species, suggesting instead that UV colour is under natural selection.”

L251: ‘Create luminance channel function by averaging together the long- and medium wavelength-sensitive channels, which is the standard in visual ecology’. This needs a citation and it depends on the taxa. For example, fish are thought to process luminance using their double cones, but also use double cones for processing chromatic information. Whereas birds have single cones for colour vision but are thought to only process luminance information through double cones. So it is a little more complicated than stated.

Revised as suggested, Lines 377-379: “For cone-catch models that did not include a luminance channel for edge intensity analysis (dog, human, and snake), we created one using the ‘Create luminance channel’ function by averaging together photoreceptor channels following the literature (see Supplementary Table 1).”

We have also added λ_{\max} and Weber fraction information for each receiver to this new Supplementary Table 1, and added an additional section “Additional Quantitative Color Pattern Analysis (QCPA) parameters” to Supplementary Text.

L254: There are multiple ways of measuring contrast, so this needs to be explained in more detail here, rather than rely on referencing the QCPA paper or the supplementary information. This doesn’t tell you how these measures are calculated, which is important for the results. For example, in Figure 2, it needs to be made clear what the units refer to.

Revised as suggested, we have now clarified units in both Fig. 3 and 4 (originally Fig. 2-3), and added text to the Methods:

Revised lines 366-371: “To directly assess the perceived chromatic contrast of entire snake colour patterns, we used the mean chromatic colour distance (ΔS , the Euclidean distance

between points in receiver colour space) as calculated by the LEIA analysis on the RNL-clustered image. ΔS summarizes the degree of colour contrast as a distance metric between two patches (e.g., ΔS between two pixels of the same colour will be 0) rather than directly comparing average brightness, and mean ΔS ($\overline{\Delta S}$) represents an average of these values across the whole organism.”

Revised Lines 323-327: “Then, scores were combined into a snake’s UV reflectance index by first averaging scores across the two observers, and then multiplying the mean brightness score by the mean proportion of the body with UV reflectance (coverage). We performed these calculations separately for dorsal and ventral surface indices and summed them for an overall reflectance index per snake, and all three of these indices were retained for downstream statistical analysis.”

Again, many thanks to all reviewers for their thoughtful comments on this work!

REVIEWERS' COMMENTS

Reviewer #1 (Remarks to the Author):

The authors have done a thorough revision of their manuscript, and I feel that it is much stronger as a result of this. I have a few remaining comments:

L374-L376: The reference to average brightness in this explanation is confusing. If you are using ΔS for colour contrast measurements, this is the difference in chromaticity of two colour patches, and is not a measure of brightness. The example of a Zebra and a grey horse is measuring difference in brightness, as the colour contrast for black and white would be 0. The discussion between colour contrast and brightness contrast needs disentangling as it does not make sense as it is currently written. A better example would be how blue and yellow would combine to make green.

Figure 4a: Can the authors please specify which ocular filters/ oil droplets were considered when calculating the spectral sensitivity curves for each species? As mentioned in the MICA toolbox these should be considered and specified when using the data for visual modelling. Examining the shape of the curves, all the UV – or UV weak species appear to have the relevant opsin spectral sensitivity curves multiplied by the ocular media or oil droplets, except for the dog. I would find it unlikely that dogs do not have some kind of UV blocking properties in their cornea etc that would limit the transmission of UV to the retina and this reduce the Beta curve that are shown at 360nm. Can the authors provide a reference to show that these do or do not exist?

L111 onwards: It should be mentioned that the relationship between ΔS and conspicuousness may not be linear, as shown in:

Santiago, C., Green, N. F., Hamilton, N., Endler, J. A., Osorio, D. C., Marshall, N. J. and Cheney, K. L. (2020). Does conspicuousness scale linearly with colour distance? A test using reef fish.

Proceedings of the Royal Society B: Biological Sciences, 287 (1935) 20201456. doi: 10.1098/rspb.2020.1456

Reviewer #2 (Remarks to the Author):

Dear Authors,

Overall I am impressed by the attention given to revisions, which have really improved the manuscript in my view. I have a few thoughts and pieces of feedback remaining, some of them new following revisions:

I understand the investigation of colour production mechanisms is likely the next big challenge in this line of research, but providing some initial information on your hypothesis that UV reflection is structural and can overlay pigmentary colours, would be very useful. This could draw on comparative evidence from lizards, or if you are able to push the boat out and include a SEM image, that would be amazing.

I still do not think "(all Pagel's $\lambda < 0.2$, and all $p > 0.1$)" is reported right. The tests of Pagel's λ are whether λ differs from zero or one. As λ is low, I assume the test reported is that it is not significantly different from zero, but this should be reported " $p(\neq 0) =$ " as noted in my previous review.

You are absolutely right that we lack good quality information on background reflectance in natural habitat. I really would have hoped someone would have been funded to take a comprehensive look at this by now! I had not seen the Guzmán et al. paper measuring bark and lichen reflectance in Costa Rica and looked it up. The authors have low confidence in their UV measurements, and do not report these in their main text ("After careful consideration, bands at the edges of the spectrometer range (the shortest and longest wavelengths) were omitted due to relatively low signal-to-noise ratios of these wavelength regions. As a result, the wavelength range of 450–1000

nm (171 bands) was retained for all the ensuing data analysis.”). Thus, I am not aware of studies that report high UV bark reflection – data is limited, but most reports are of low bark UV reflection, or limited UV contrast with non UV reflecting stimuli – e.g.:

Campbell SA, Borden JH. Bark reflectance spectra of conifers and angiosperms: implications for host discrimination by coniferophagous bark and timber beetles. *The Canadian Entomologist*. 2005;137(6):719-722. doi:10.4039/n04-082

It is however a good point that lichens can be UV reflective – there is better data of this e.g. <https://doi.org/10.1016/j.rse.2020.111955> and the Majerus paper cited.

L99 (track changes version) Doesn't this now suggest they are coupled?

L173 “other ecological factors that can covary with habitat usage” I think these need to be indicated.

L444 “We note that computer-scored metric required the highest quality photos in raw format, so these represent a subset of ~30% of all snakes measured and a concomitant reduction in statistical power. (N = 65 species).” I think the main methods need to be clearer that it was not possible to create mspecs for some individuals because of animal movement & consequent impossibility of image alignment.

Snake images throughout – why have the images not been standardised to the reflectance standard (some are pink, or various shades of grey)? Doing so would facilitate comparison.

The SI does a good job of addressing the concerns raised by myself and other reviewers about the possibilities for scale thickness, shed state, oils, shine, and selection on non-UV colour to explain results. I think the section reads a bit too much like a ‘response to reviewer comments’ at the moment – which it is, but I suggest editing the tone to incorporate these as valid possibilities to be evaluated (at the moment it reads a bit too much like things to be ruled out). It still seems quite plausible to me that juveniles are found to be more UV reflective than adults in the analysis because they shed more frequently, or are oilier, rather than natural selection on UV colour.

“The additional UV reflectance that we have documented to these otherwise broadly-visible colour patterns should not be assumed a priori to function differently “ UV/white colour/countershading – worth considering that background matching countershading of snakes viewed from below (which may happen for arboreal snakes) would need to reflect UV to better match the UV in downwelling light (difficult to be reflective enough though – looking up at arboreal snakes they are generally silhouetted).

Yours sincerely,

Will Allen

Revised as suggested; note that our major revision was to remove mention of the data from the literature search from the introduction as requested, and instead detail these results as a new first subsection of Results.

Reviewer #1 (Remarks to the Author):

The authors have done a thorough revision of their manuscript, and I feel that it is much stronger as a result of this. I have a few remaining comments:

- 1.) L374-L376: The reference to average brightness in this explanation is confusing. If you are using ΔS for colour contrast measurements, this is the difference in chromaticity of two colour patches, and is not a measure of brightness. The example of a Zebra and a grey horse is measuring difference in brightness, as the colour contrast for black and white would be 0. The discussion between colour contrast and brightness contrast needs disentangling as it does not make sense as it is currently written. A better example would be how blue and yellow would combine to make green.

Revised as suggested (new manuscript line 399-404): “Chromatic ΔS summarises the degree of colour contrast as a distance metric between two patches (*e.g.*, ΔS between two pixels of the same colour will be 0) rather than directly comparing average chromaticity, and mean chromatic ΔS represents an average of these values across the whole organism. For example, the blue, red, and yellow plumage of a scarlet macaw would have a higher chromatic ΔS than a uniformly green bird, even if they might have the same average chromaticity.”

- 2.) Figure 4a: Can the authors please specify which ocular filters/ oil droplets were considered when calculating the spectral sensitivity curves for each species? As mentioned in the MICA toolbox these should be considered and specified when using the data for visual modelling. Examining the shape of the curves, all the UV – or UV weak species appear to have the relevant opsin spectral sensitivity curves multiplied by the ocular media or oil droplets, except for the dog. I would find it unlikely that dogs do not have some kind of UV blocking properties in their cornea etc that would limit the transmission of UV to the retina and this reduce the Beta curve that are shown at 360nm. Can the authors provide a reference to show that these do or do not exist?

Revised as suggested: We have added a sentence to the caption of figure 4 referring the reader to Supplementary Table 1, where we have added a column and citations for UV transmission of ocular media/oil droplets for each of the receivers in our analysis to that table.

Revised manuscript, Fig 4 caption: “See Supplementary Table 1 for details and citations of visual systems used, including λ_{max} , oil droplets/ocular media affecting UV transmission, and calculation of luminance channels.”

3.) L111 onwards: It should be mentioned that the relationship between deltaS and conspicuousness may not be linear, as shown in:

Santiago, C., Green, N. F., Hamilton, N., Endler, J. A., Osorio, D. C., Marshall, N. J. and Cheney, K. L. (2020). Does conspicuousness scale linearly with colour distance? A test using reef fish. *Proceedings of the Royal Society B: Biological Sciences*, 287 (1935) 20201456. doi: 10.1098/rspb.2020.1456

Revised as suggested with new citation included (new manuscript line 147-150).

Reviewer #2 (Remarks to the Author):

Dear Authors,

Overall I am impressed by the attention given to revisions, which have really improved the manuscript in my view. I have a few thoughts and pieces of feedback remaining, some of them new following revisions:

1.) I understand the investigation of colour production mechanisms is likely the next big challenge in this line of research, but providing some initial information on your hypothesis that UV reflection is structural and can overlay pigmentary colours, would be very useful. This could draw on comparative evidence from lizards, or if you are able to push the boat out and include a SEM image, that would be amazing.

As a manuscript on the ecological relevance of colouration that did not take any direct data on colour production, we are hesitant to overspeculate about potential cellular mechanisms in the main text. We appreciate that this may mean that our supplementary discussion may be overlooked by some readers who only read the

main text, but we think that this is preferable to prioritising conclusions about data we did not collect over data that we did present.

- 2.) I still do not think “(all Pagel’s $\lambda < 0.2$, and all $p > 0.1$)” is reported right. The tests of Pagel’s λ are whether λ differs from zero or one. As λ is low, I assume the test reported is that it is not significantly different from zero, but this should be reported “ $p(\neq 0) =$ ” as noted in my previous review.

Revised to meet the intent of the comment without introducing unnecessary confusion, lines 94-97: “UV reflectance on both dorsal and ventral surfaces showed little phylogenetic signal, even when accounting for phylogenetic uncertainty (all Pagel’s $\lambda < 0.2$ and all $p > 0.1$, and thus not significantly different from zero).”

- 3.) You are absolutely right that we lack good quality information on background reflectance in natural habitat. I really would have hoped someone would have been funded to take a comprehensive look at this by now! I had not seen the Guzmán et al. paper measuring bark and lichen reflectance in Costa Rica and looked it up. The authors have low confidence in their UV measurements, and do not report these in their main text (“After careful consideration, bands at the edges of the spectrometer range (the shortest and longest wavelengths) were omitted due to relatively low signal-to-noise ratios of these wavelength regions. As a result, the wavelength range of 450–1000 nm (171 bands) was retained for all the ensuing data analysis.”). Thus, I am not aware of studies that report high UV bark reflection – data is limited, but most reports are of low bark UV reflection, or limited UV contrast with non UV reflecting stimuli – e.g.:

Campbell SA, Borden JH. Bark reflectance spectra of conifers and angiosperms: implications for host discrimination by coniferophagous bark and timber beetles. *The Canadian Entomologist*. 2005;137(6):719-722. doi:10.4039/n04-082

It is however a good point that lichens can be UV reflective – there is better data of this e.g. <https://doi.org/10.1016/j.rse.2020.111955> and the Majerus paper cited.

After further reviewing the suggested article on lichens, we have decided not to include it in our literature cited list as its focus is on ground lichens in boreal forests while our paper explores hypotheses behind the higher UV reflectance in arboreal snakes in primarily tropical habitats.

- 4.) L99 (track changes version) Doesn’t this now suggest they are coupled?

Thank you for catching this typo; the sentence was supposed to read “...UV reflectance could not be reliably predicted...” rather than “...could be...” We have edited the sentence to reflect this change (new manuscript lines 99-103).

5.) L173 “other ecological factors that can covary with habitat usage” I think these need to be indicated.

Revised as suggested (new manuscript lines 168-170). This sentence now includes an example of one such ecological factor (diet), with a new reference.

6.) L444 “We note that computer-scored metric required the highest quality photos in raw format, so these represent a subset of ~30% of all snakes measured and a concomitant reduction in statistical power. (N = 65 species).” I think the main methods need to be clearer that it was not possible to create mspecs for some individuals because of animal movement & consequent impossibility of image alignment.

Revised as suggested (new manuscript lines 420-424): “We note that the computer-scored metric required the highest quality photos in raw format, in which the visible and UV photos of the specimen could be perfectly aligned, which was not always possible (*e.g.*, if the specimen was moved slightly in between photos). Usable image pairs represent a subset of ~30% of all snakes measured and a concomitant reduction in statistical power (N = 65 species).”

7.) Snake images throughout – why have the images not been standardised to the reflectance standard (some are pink, or various shades of grey)? Doing so would facilitate comparison.

Snake images have been rebalanced and standardised as suggested.

8.) The SI does a good job of addressing the concerns raised by myself and other reviewers about the possibilities for scale thickness, shed state, oils, shine, and selection on non-UV colour to explain results. I think the section reads a bit too much like a ‘response to reviewer comments’ at the moment – which it is, but I suggest editing the tone to incorporate these as valid possibilities to be evaluated (at the moment it reads a bit too much like things to be ruled out). It still seems quite plausible to me that juveniles are found to be more UV reflective than adults in the analysis because they shed more frequently, or are oilier, rather than natural selection on UV colour.

We have extensively revised this section as suggested. We changed the overall tone to present these possibilities more so as valid ideas to be considered when explaining the mechanisms behind intraspecific differences in UV reflectance rather than factors to be ruled out completely.

9.) “The additional UV reflectance that we have documented to these otherwise broadly-visible colour patterns should not be assumed a priori to function differently “
UV/white colour/countershading – worth considering that background matching
countershading of snakes viewed from below (which may happen for arboreal snakes)
would need to reflect UV to better match the UV in downwelling light (difficult to be
reflective enough though – looking up at arboreal snakes they are generally
silhouetted).

Revised as suggested (new manuscript lines 190-193): “In addition, higher amounts of UV reflectance on the ventral surfaces of these snakes could aid in crypsis when viewed from below by terrestrial or understory-dwelling predators, as these colours would presumably contrast less with the UV-intense backdrop of the sky.”

Yours sincerely,

Will Allen

Reviewer #2 (Remarks on code availability):

I had a look and everything seems in order. I don't have time to run the code though. The images could be a good resource for other studies.